# High-dimensional (Group) Adversarial Training in Linear Regression

**Yiling Xie**
School of Industrial and Systems Engineering
Georgia Institute of Technology
Atlanta, Georgia, USA
`yxie350@gatech.edu`

**Xiaoming Huo**
School of Industrial and Systems Engineering
Georgia Institute of Technology
Atlanta, Georgia, USA
`huo@gatech.edu`

## Abstract

Adversarial training can achieve robustness against adversarial perturbations and has been widely used in machine-learning models. This paper delivers a non-asymptotic consistency analysis of the adversarial training procedure under $\ell_\infty$-perturbation in high-dimensional linear regression. It will be shown that, under the restricted eigenvalue condition, the associated convergence rate of prediction error can achieve the *minimax* rate up to a logarithmic factor in the high-dimensional linear regression on the class of sparse parameters. Additionally, the group adversarial training procedure is analyzed. Compared with classic adversarial training, it will be proved that the group adversarial training procedure enjoys a better prediction error upper bound under certain group-sparsity patterns.

## 1 Introduction

Adversarial training is proposed to hedge against adversarial perturbations and has attracted much research interest in recent years. Adversarial training has been widely used in Large Language Models [14, 24], computer vision [13], cybersecurity [33], etc. While the empirical risk minimization procedure optimizes the empirical loss, the adversarial training procedure seeks conservative solutions that optimize the worst-case loss under a given magnitude of perturbation. People have actively investigated model modifications and algorithmic frameworks to improve performance and training efficiency for adversarial training under different problem settings [1, 10, 12, 22, 23, 27, 29].

We are interested in understanding the fundamental properties of adversarial training from a statistical viewpoint. A standard approach for statisticians to evaluate statistical or machine-learning models is to investigate whether the estimator obtained from the model can achieve the minimax rate [25]. In this paper, we will give the non-asymptotic convergence rate of the prediction error in high-dimensional adversarial training. The associated convergence rate achieves the minimax rate under certain settings, which will be clarified in Section 2.2.

In machine-learning models, adversarial training has the following mathematical formulation:

$$\min_\beta \frac{1}{n} \sum_{i=1}^n \sup_{\|\Delta\| \le \delta} L\left(\boldsymbol{X}_i + \Delta, Y_i, \beta\right),$$

where $(\boldsymbol{X}_1, Y_1), ..., (\boldsymbol{X}_n, Y_n)$ are given samples, $\Delta$ is the perturbation, $\|\cdot\|$ is the perturbation norm, $\delta$ is the perturbation magnitude, $\beta$ is the model parameter, and $L(\boldsymbol{x}, y, \beta)$ is the loss function with $\boldsymbol{x}$ being the input variable and $y$ being the response variable.

Regarding the choice of the perturbation norm, we focus on $\ell_\infty$-perturbation, i.e., $\|\Delta\| = \|\Delta\|_\infty$. Some literature has pointed out that $\ell_\infty$-perturbation could help recover the model sparsity [21, 28].

38th Conference on Neural Information Processing Systems (NeurIPS 2024).

For example, [28] has proved that the asymptotic distribution of adversarial training estimator under $\ell_\infty$-perturbation has a positive mass at $0$ when the underlying parameter is $0$. Since the sparsity assumption could improve the model interpretation and reduce problem complexity [11], especially in high-dimensional regimes, $\ell_\infty$-perturbation will be studied, and certain sparsity patterns will be assumed in this paper. In terms of the loss function, we focus on the loss in the linear regression, i.e., $L(\boldsymbol{x}, y, \beta) = (\boldsymbol{x}^\top \beta - y)^2$. In particular, many of the existing theoretical explorations on adversarial training are based on linear models [15, 16, 21, 28, 30, 31], which admits advanced analytical analysis and sheds light on the characteristics of adversarial training in more general settings and applications. In this regard, the linear regression is considered in this paper. In short, we will focus on the adversarial-trained linear regression as shown in the following:

$$\widehat{\beta} \in \arg\min_\beta \frac{1}{n} \sum_{i=1}^n \max_{\|\Delta\|_\infty \leq \delta} \left( (\boldsymbol{X}_i + \Delta)^\top \beta - Y_i \right)^2, \tag{1}$$

where $\boldsymbol{X}_i, \beta \in \mathbb{R}^p$ and $Y_i \in \mathbb{R}$. For the convenience of analysis, we write the given $n$ samples $(\boldsymbol{X}_1, Y_1), ..., (\boldsymbol{X}_n, Y_n)$ in the matrix form: $\boldsymbol{X} = (\boldsymbol{X}_1, ..., \boldsymbol{X}_n)^\top \in \mathbb{R}^{n \times p}$ and $\boldsymbol{Y} = (Y_1, ..., Y_n)^\top \in \mathbb{R}^{n \times 1}$, where we call $\boldsymbol{X}$ the design matrix.

This paper delivers the convergence analysis under the high-dimension setting, where we suppose the dimension of the model parameter $\beta$ is larger than the sample size, i.e., $p > n$. Further, the parameter sparsity is assumed. Specifically, we suppose that only a subset of the elements of the $p$-dimensional ground-truth parameter $\beta_*$ is nonzero. If the size of the nonzero subset is $s$, it will be shown that the resulting prediction error of problem (1), i.e., $\|\boldsymbol{X}(\widehat{\beta} - \beta_*)\|_2^2 / n$, is of the order $s \log p / n$ under the restricted eigenvalue condition. The restricted eigenvalue condition is a standard assumption in the literature of sparse high-dimensional linear regression [2, 3, 4, 17]. Notably, the rate $s \log p / n$ is optimal in the minimax sense, up to a logarithmic factor, for all estimators over a class of $s$-sparse $p$-dimensional vectors if there are $n$ training samples [20].

Our aforementioned results have the following implications. Firstly, in addition to robustness towards perturbation, our results show that adversarial training is beneficial regarding statistical optimality. This means the resulting estimator can achieve the minimax convergence rate for prediction error. To the best of our knowledge, we are the first to prove the minimax optimality of the adversarial training estimator. Secondly, our analysis illustrates that the $\ell_\infty$-perturbation is recommended if the sparsity condition, i.e., the ground truth $\beta_*$ is supported on a subset of $\{1, ..., p\}$, is known and a fast theoretical error convergence rate is required.

The convergence rate of the group adversarial training is also investigated. In the literature, the group effect has been studied in (finite-type) Wasserstein distributionally robust optimization problems [5, 7]. Since adversarial training is equivalent to $\infty$-type Wasserstein distributionally robust optimization problem [9], the formulation of group Wasserstein distributionally robust optimization problem discussed in [5, 7] can be generalized to the adversarial training problem. We give a formal formulation of the group adversarial training problem based on frameworks in [5, 7]. Further, we derive the non-asymptotic convergence rate of the prediction error for the group adversarial training problem. It will be shown that group adversarial training can achieve a faster convergence upper bound if certain group sparsity structures are satisfied. The details can be found in Section 3.2.

## 1.1 Related Work

We review and compare some related work in this subsection.

The asymptotic behavior of $\ell_\infty$-adversarial training estimator in the generalized linear model has been discussed in [28]. Notably, the paper [28] studies the behavior of adversarial training estimator from an asymptotic point of view, while our paper delivers a non-asymptotic analysis. More specifically, analysis in [28] is based on the asymptotic distribution of the adversarial training estimator, while our work is to give a non-asymptotic upper bound of the prediction error of the adversarial training estimator. More discussions can be found in Remark 2.7.

The prediction error of $\ell_\infty$-adversarial training estimator has been briefly analyzed in [21], where the proven convergence is of the order $1/\sqrt{n}$ in terms of $n$. The results in our paper are different in the following two perspectives. Firstly, a faster convergence rate of the order $1/n$ in terms of $n$ is given, and the associated rate is minimax optimal up to a logarithmic factor. Secondly, we have incorporated the sparsity setting in the model analysis, while no sparsity pattern is considered in

theoretical analysis for $\ell_\infty$-adversarial training in [21]. More discussions can be found in Remark 2.8.

The paper [30] also investigates the convergence of adversarial training estimator in linear regression. The derivations in [30] are based on the assumption that the input variable $X$ follows $p$-dimensional Gaussian distribution while our analysis imposes the restricted eigenvalue condition. In addition, notice that [30] argues the superiority of incorporating the sparsity information by deriving lower bounds for the estimator error while we directly prove the rate optimality of the adversarial training estimator under the sparsity assumption. Also, [30] applies $\ell_2$-perturbation while our work focuses on $\ell_\infty$-perturbation.

In the literature, it has been proven that multiple estimators, including LASSO, Dantzig selector, and square-root LASSO, can achieve the minimax rate (up to a logarithmic factor) in high-dimensional sparse linear regression [3, 4, 11, 20]. However, to the best of our knowledge, no literature has investigated this property for the widely used adversarial training model. We are the first to study whether the adversarial training estimator can be minimax optimal, and our theoretical analysis implies that the answer is yes, i.e., the adversarial training estimator under $\ell_\infty$-perturbation enjoys rate optimality. In addition, the group lasso has been intensely studied to explore the parameter group structure [17, 32] while group adversarial training imposes group structure on the perturbation. It will be shown that the group adversarial training estimator shares a similar convergence rate with the group LASSO estimator. Our proof technique is developed upon and extends the technical methods in the aforementioned papers [3, 4, 11, 17].

## 1.2 Notations and Preliminaries

We introduce some notations, which will be used in the rest of the paper. For vector $z \in \mathbb{R}^p$, we use $\|z\|_q$ to denote the $\ell_q$ norm of the vector $z$, i.e., $\|z\|_q^q = \sum_{j=1}^p |z_j|^q, 1 \leq q < \infty, \|z\|_\infty = \max_{1 \leq j \leq p} |z_j|$. We use $e_j \in \mathbb{R}^p, 1 \leq j \leq p$, to denote the basis vectors where the $j$th component is 1 and 0 otherwise. $I_n \in \mathbb{R}^{n \times n}$ denotes the identity matrix. For some set $S$, we use $S^c$ to denote the complement set of $S$ and $|S|$ to denote the cardinality of $S$. If the set $S$ is the subset of $\{1, ..., p\}$, we use $z_S \in \mathbb{R}^{|S|}$ to denote the subvector indexed by elements of $S$.

We clarify some preliminary settings that will be used in this paper. Throughout this paper, we suppose the high-dimension setting holds, the samples are generated from the Gaussian linear model, and the design matrix is normalized. We conclude these conditions with the following assumptions:

**Assumption 1.1** (High-dimension)**.** *The parameter dimension $p$ is larger than the sample size $n$, i.e., we have $n < p$.*

**Assumption 1.2** (Gaussian linear model)**.** *The design matrix $X$ is fixed and the response vector $Y$ is generated by the following: $Y = X\beta_* + \epsilon$, where $\epsilon$ has i.i.d. entries $\mathcal{N}(0, \sigma^2)$.*

**Assumption 1.3** (Normalization)**.** *The design matrix $X$ is normalized such that $\|Xe_j\|_2 \leq \sqrt{n}$, for $1 \leq j \leq p$.*

## 1.3 Organization of this Paper

The remainder of this paper is organized as follows. In Section 2, we derive the convergence rate of the adversarial training estimator in high-dimensional linear regression. In Section 3, we derive the convergence rate of group adversarial training and compare it with the existing adversarial training. Numerical experiments are conducted in Section 4. Possible future work is discussed in Section 5. The proofs are relegated to the Appendix whenever possible.

## 2 $\ell_\infty$-Adversarial-Trained Linear Regression

In this section, we will first introduce the problem formulation of the adversarial training in linear regression under $\ell_\infty$-perturbation and then deliver the convergence analysis of the prediction error under $\ell_\infty$-perturbation in the high-dimensional setting.

## 2.1 Problem Formulation

In this subsection, we give the problem formulation of $\ell_\infty$-adversarial-trained linear regression and discuss its dual.

Recall that the $\ell_\infty$-adversarial training problem in linear regression has the formulation shown in (1). The solution $\widehat{\beta}$ to the optimization problem (1) is used to estimate the ground-truth data generating parameter $\beta_*$, seeing Assumption 1.2. In the inner optimization problem, we compute the worst-case square loss between the response variable and the linear prediction among the perturbations. The perturbations are added to the input variable and with the largest $\ell_\infty$-norm $\delta$. In the outer optimization problem, we optimize the empirical expectation of the worst-case loss of given samples.

The optimization problem (1) can be further simplified by considering its dual formulation, which is shown as follows [21, Proposition 1].

**Proposition 2.1** (Dual Formulation of problem (1)). *If we denote the optimal value of problem* (1) *by* $R(\delta)$*, then we have that*

$$R(\delta) = \min_\beta \frac{1}{n} \sum_{i=1}^n \left( |\boldsymbol{X}_i^\top \beta - Y_i| + \delta\|\beta\|_1 \right)^2. \tag{2}$$

We discuss the advantages and theoretical insights we could get by considering the dual problem (2). Note that the dual formulation (2) removes the inner maximization of problem (1), and the associated objective function is a convex function of $\beta$. Thus, it will be more convenient to solve the dual problem (2) than the primal problem (1). Also, the expansion of the objective function in (2) yields the following:

$$\min_\beta \frac{1}{n} \sum_{i=1}^n (\boldsymbol{X}_i^\top \beta - Y_i)^2 + \delta\|\beta\|_1 \left( \frac{2}{n} \sum_{i=1}^n |\boldsymbol{X}_i^\top \beta - Y_i| \right) + \delta^2\|\beta\|_1^2, \tag{3}$$

where the residual term $\delta^2\|\beta\|_1^2$ will be of high order if we let $\delta$, for example, be proportional to the inverse of a positive power of $n$. Regardless of the high order residual term, the objective function in problem (2) can be viewed as the sum of the loss function in linear regression and a regularization term depending on $\|\beta\|_1$. This implies that $\ell_\infty$-adversarial-trained linear regression has a regularization effect. We refer to [21, 28] and references therein for more discussions about the regularization effect of adversarial training. Since the well-known LASSO is formulated by imposing the $\ell_1$-norm regularization term and enjoys the minimax convergence rate of the prediction error [20], the dual formulation (2) of $\ell_\infty$-adversarial training in linear regression and its expansion (3) may indicate a fast convergence of its prediction error for the adversarial training estimator.

## 2.2 Convergence Analysis

In this subsection, we will first introduce the restricted eigenvalue condition and then derive the convergence rate of the prediction error for the adversarial training estimator in high-dimensional linear regression under the restricted eigenvalue condition and $\ell_\infty$-perturbation. We will also discuss the high-probability arguments upon which we prove the optimality of the associated adversarial training estimator.

Before we deliver the convergence analysis, we make the following assumption.

**Assumption 2.2** (Restricted Eigenvalue Condition). *The matrix* $\boldsymbol{X} \in \mathbb{R}^{n \times p}$ *satisfies the restricted eigenvalue condition if there exists a positive number* $\gamma = \gamma(s, c_1) > 0$ *such that*

$$\min\left\{ \frac{\|\boldsymbol{X}v\|_2}{\sqrt{n}\|v\|_2} : |S| \leq s, v \in \mathbb{R}^p\backslash\{\boldsymbol{0}\}, \|v_{S^c}\|_1 \leq c_1\|v_S\|_1 \right\} \geq \gamma,$$

*where* $S$ *is some subset of* $\{1, ..., p\}$*.*

In the sequel, we use the notation $\mathrm{RE}(s, c_1)$ to denote the restricted eigenvalue condition w.r.t. the cardinality $s$ of the index set $S$ and the constant $c_1$ in the constrained cone, i.e., $\|v_{S^c}\|_1 \leq c_1\|v_S\|_1$. The restricted eigenvalue condition can be considered as a relaxation of the positive semidefiniteness of the gram matrix $\boldsymbol{X}^\top \boldsymbol{X}$ and is a useful technique in theoretical analysis in the sparse high-dimensional analysis [11].

Equipped with Assumption 2.2, we have the following convergence result of prediction error.

**Theorem 2.3** (Prediction Error Analysis for Adversarial Training). *Suppose the adversarial training problem* (1) *satisfies*

$$\frac{2\|\boldsymbol{X}^\top \boldsymbol{\epsilon}\|_\infty}{\|\boldsymbol{\epsilon}\|_1} \leq \delta. \tag{4}$$

*If $\beta_*$ is supported on a subset $S$ of $\{1, ..., p\}$ where $|S| \leq s$, and the design matrix $\boldsymbol{X}$ satisfies* $\mathrm{RE}(s, 3)$ *with parameter $\gamma(s, 3)$, then we have that*

$$\frac{1}{2n}\|\boldsymbol{X}(\widehat{\beta} - \beta_*)\|_2^2 \leq 3\delta^2 s \max\left\{\frac{9}{\gamma^2(s, 3)}\left(\frac{\|\boldsymbol{\epsilon}\|_1}{n}\right)^2, 164\|\beta_*\|_2^2\right\}$$

Theorem 2.3 shows that the upper bound of the prediction error mainly depends on the sparsity cardinality $s$ of the ground-truth parameter $\beta_*$ and perturbation magnitude $\delta$. The perturbation magnitude $\delta$ is assumed to be equal to or larger than $2\|\boldsymbol{X}^\top \boldsymbol{\epsilon}\|_\infty/\|\boldsymbol{\epsilon}\|_1$. We could apply the concentration inequalities to give a closed-form expression of the perturbation magnitude, based on which the convergence rate of the prediction error is derived. The convergence rate holds with a high probability and can be found in the following corollary.

**Corollary 2.4.** *Consider the adversarial training problem* (1) *with perturbation magnitude*

$$\delta = \frac{4}{\sqrt{\frac{2}{\pi} - \frac{1}{10}}}\sqrt{\frac{\log p}{n}}. \tag{5}$$

*If $\beta_*$ is supported on a subset $S$ of $\{1, ..., p\}$ where $|S| \leq s$, and the design matrix $\boldsymbol{X}$ satisfies* $\mathrm{RE}(s, 3)$ *with parameter $\gamma(s, 3)$, then we have that*

$$\frac{1}{2n}\|\boldsymbol{X}(\widehat{\beta} - \beta_*)\|_2^2 \leq 192\frac{s \log p}{n}\max\left\{\frac{9}{\gamma^2(s, 3)}\left(\frac{\|\boldsymbol{\epsilon}\|_1}{n}\right)^2, 164\|\beta_*\|_2^2\right\} \tag{6}$$

*holds with a probability greater than $1 - 2\exp(-C_1 n) - 2/p$, where $C_1$ is a positive constant.*

**Remark 2.5.** *We discuss the choice of $\delta$. Corollary 2.4 implies that the perturbation magnitude $\delta$ should be of the order $1/\sqrt{n}$ in order to derive the non-asymptotic convergence rate* (6). *The associated order is consistent with the asymptotic analysis in [28], where the sparsity-recovery ability could be proven in the asymptotic sense if the sample size is of the order $1/\sqrt{n}$.*

In Corollary 2.4, we choose $\delta$ as is shown in (5). Under this setting, it can be proven that the inequality (4) holds with a high probability. Then, we adopt Theorem 2.3 and could have the expression of the prediction error in terms of $p$ and $n$ as shown in (6).

The convergence rate could be further simplified in the following corollary if both the error variance $\sigma^2$ and the $\ell_2$-norm of the ground-truth parameter $\beta_*$ are bounded.

**Corollary 2.6.** *Under the assumptions stated in Corollary 2.4, suppose there exists a finite positive constant $R$ such that*

$$2\sqrt{41}\|\beta_*\|_2 \leq R, \quad \sigma < \frac{1}{6}\gamma(s, 3)R,$$

*then we have that*

$$\frac{1}{2n}\|\boldsymbol{X}(\widehat{\beta} - \beta_*)\|_2^2 \leq 192\frac{s \log p}{n}R^2$$

*holds with a probability greater than $1 - 2/p - 2\exp(-C_1 n) - \exp(-n)$, where $C_1$ is a positive constant.*

**Remark 2.7.** *Corollary 2.6 investigates the behavior of the adversarial training estimator under $\ell_\infty$-perturbation by computing the resulting prediction error while [28] studies the behavior of the adversarial training estimator under $\ell_\infty$-perturbation by deriving the associated limiting distribution. Both [28] and our results consider the sparsity condition. [28] proves that $\ell_\infty$-adversarial training can help recover sparsity asymptotically if the parameter sparsity is known while our paper, i.e., Corollary 2.6, provides a fast non-asymptotic convergence rate for prediction error under the sparsity condition.*

**Remark 2.8.** *Corollary 2.6 illustrates that the convergence rate of prediction error for $\ell_\infty$-adversarial training in linear regression is of the order $s \log p/n$ while the prediction error shown in [21] has a lower order $1/\sqrt{n}$ in terms of $n$. Our paper achieves a faster rate by incorporating the sparsity information and applying the restricted eigenvalue condition.*

Corollary 2.6 implies that the prediction error of high-dimensional $\ell_\infty$-adversarial-trained estimator is of the order $s \log p/n$. This order is optimal up to a logarithmic factor in the minimax sense for any estimators over a class of $s$-sparse vectors in $\mathbb{R}^p$ when $n$ samples are given [2, 20].

## 3 Group Adversarial Training

This section will elaborate on the formulation of group adversarial training and the associated convergence rate. Also, we compare group adversarial training under $(2, \infty)$-perturbation with classic adversarial training under $\ell_\infty$-perturbation.

Since the adversarial training forces the perturbation with uniform magnitude to each component of the input variable, it may not perform well if the input variable has a group effect. The group structure exists in many real-world problems. For example, groups of genes act together in pathways in gene-expression arrays [18], and financial data can be grouped by different sectors and industries to help market prediction [6]. Also, if an input variable is a multilevel factor and dummy variables are introduced, then these dummy variables act in a group [32]. Group adversarial training can tackle the group effect by adding group-structured perturbation. The detailed formulation can be seen in Section 3.1.

### 3.1 Problem Formulation

In this subsection, we describe the formulation of the group adversarial training.

Suppose the input variable $x$ can be divided into $L$ non-overlapped groups. Then, we have the definition of the group-structured weighted norm accordingly in the following proposition, where the associated dual norm is also stated.

**Proposition 3.1** (Proposition 5 in [5], Theorem 2.2 in [7]). *Consider a vector $\boldsymbol{x} = (\boldsymbol{x}^1, ..., \boldsymbol{x}^L)$, where each $\boldsymbol{x}^l \in \mathbb{R}^{p_l}$, and $\sum_{l=1}^L p_l = p$. Define the weighted $(r, s)$-norm of $\boldsymbol{x}$ with the $L$-dimensional weight vector $\boldsymbol{\omega} = (\omega_1, ..., \omega_L)$ to be:*

$$\|\boldsymbol{x}_{\boldsymbol{\omega}}\|_{r,s} = \left( \sum_{l=1}^L \|\omega_l \boldsymbol{x}^l\|_r^s \right)^{1/s}, 1 \le s < \infty,$$

$$\|\boldsymbol{x}_{\boldsymbol{\omega}}\|_{r,\infty} = \max_{1 \le l \le L} \|\omega_l \boldsymbol{x}^l\|_r, \quad s = \infty,$$

*where $\omega_l > 0, \forall l$ and $r \ge 1$. Then, the dual norm of $(r, s)$-norm with weight $\boldsymbol{\omega}$ is the $(q, t)$-norm with weight $\boldsymbol{\omega}^{-1} = (1/\omega_1, ..., 1/\omega_L)$, i.e. $\|\boldsymbol{x}_{\boldsymbol{\omega}^{-1}}\|_{q,t}$, where $1/r + 1/q = 1$ and $1/s + 1/t = 1$.*

To handle the group structure in the input variable, the weighted $(r, s)$-norm is applied to add group structure in the perturbation accordingly, and the group adversarial training is formulated as follows,

$$\min_\beta \frac{1}{n} \sum_{i=1}^n \sup_{\|\Delta_{\boldsymbol{\omega}}\|_{r,s} \le \delta} \left( (\boldsymbol{X}_i + \Delta)^\top \beta - Y_i \right)^2,$$

where $\boldsymbol{\omega} = (\omega_1, ..., \omega_L)$.

Recall we focus on adversarial training problems under $\ell_\infty$-perturbation, high-dimension setting, and sparsity condition. Under this consideration, we let $s = \infty$ and $r = 2$, and then the associated group adversarial training problem under $(2, \infty)$-perturbation has the following expression:

$$\min_\beta \frac{1}{n} \sum_{i=1}^n \sup_{\|\Delta_{\boldsymbol{\omega}}\|_{2,\infty} \le \delta} \left( (\boldsymbol{X}_i + \Delta)^\top \beta - Y_i \right)^2. \tag{7}$$

To facilitate convenience for the computation and analysis, similar to our study towards classic adversarial training in Section 2.1, we derive the dual formulation of problem (7) in the following proposition. One can check that the corresponding objective in the dual formulation (8) is also convex.

**Proposition 3.2** (Dual Formulation of problem (7)). *If we denote the optimal value of problem* (7) *by* $\widetilde{R}(\delta)$, *then we have that*

$$\widetilde{R}(\delta) = \min_{\beta} \frac{1}{n} \sum_{i=1}^{n} \left( |(\boldsymbol{X}_i + \Delta)^\top \beta - Y_i| + \delta \|\beta_{\boldsymbol{\omega}^{-1}}\|_{2,1} \right)^2, \tag{8}$$

*where*

$$\|\beta_{\boldsymbol{\omega}^{-1}}\|_{2,1} = \sum_{l=1}^{L} \frac{1}{\omega_l} \|\beta^l\|_2.$$

## 3.2 Convergence Analysis

In this subsection, we deliver the convergence analysis of the prediction error of the estimator obtained from group adversarial training under $(2, \infty)$-perturbation, i.e., problem (7).

First, we clarify some notations for subsequent analysis. In terms of the group structure of the input variable and the perturbation, we focus on non-overlapped cases. Assume that the index set $\{1, ..., p\}$ has the prescribed (disjoint) partition $\{1, ..., p\} = \bigcup_{l=1}^{L} G_l$. We use $p_l$ to denote the cardinality of each group, i.e., $|G_l| = p_l$.

Consider the group sparsity structure in the ground-truth parameter $\beta_* \in \mathbb{R}^p$, where sparsity is imposed at the group level instead of on individual components. Specially, the set $J \subset \{1, ..., L\}$ denotes a set of groups and $\beta_*$ is supported at these $J$ groups, i.e., $\beta_*$ is supported on the $G_J = \bigcup_{l \in J} G_l$.

We make the following assumption before we proceed to derive the convergence analysis.

**Assumption 3.3** (Group Restricted Eigenvalue Condition). *The matrix* $\boldsymbol{X} \in \mathbb{R}^{n \times p}$ *satisfies the group restricted eigenvalue condition if there exists a positive number* $\kappa = \kappa(g, c_2) > 0$ *such that*

$$\min \left\{ \frac{\|\boldsymbol{X}v\|_2}{\sqrt{n}\|v_{G_J}\|_2} : |J| \le g, v \in \mathbb{R}^p \backslash \{\boldsymbol{0}\}, \sum_{l \in J^c} \frac{1}{\omega_l} \|v^l\|_2 \le c_2 \sum_{l \in J} \frac{1}{\omega_l} \|v^l\|_2 \right\} \ge \kappa,$$

*where* $J$ *is some subset of* $\{1, ..., L\}$.

In the sequel, we use the notation $\mathrm{GRE}(g, c_2)$ to denote the restricted eigenvalue condition w.r.t. the cardinality $g$ of the index set $J$ and the constant $c_2$ in the constrained cone, i.e., $\sum_{l \in J^c} \frac{1}{\omega_l} \|v^l\|_2 \le c_2 \sum_{l \in J} \frac{1}{\omega_l} \|v^l\|_2$. Group restricted eigenvalue condition is an extension of the restricted eigenvalue condition and can be used in the theoretical analysis for the group LASSO, seeing [17].

**Theorem 3.4** (Prediction Error Analysis for Group Adversarial Training). *Consider the group adversarial training problem* (7) *satisfying*

$$\frac{2\|(\boldsymbol{X}^\top \boldsymbol{\epsilon})^l\|_2}{\|\boldsymbol{\epsilon}\|_1} \le \frac{\delta}{\omega_l}, \quad \forall l.$$

*If* $\beta_*$ *is supported on a subset* $G_J$ *of* $\{1, ..., p\}$ *where* $|J| \le g$, *and the design matrix* $\boldsymbol{X}$ *satisfies* $\mathrm{GRE}(g, 3)$ *with parameter* $\kappa(g, 3)$, *then we have that*

$$\frac{1}{2n} \|\boldsymbol{X}(\widetilde{\beta} - \beta_*)\|_2^2 \le 3\delta^2 \sum_{l \in J} \frac{1}{\omega_l^2} \max \left\{ \frac{9}{\kappa^2(g, 3)} \left( \frac{\|\boldsymbol{\epsilon}\|_1}{n} \right)^2, 164\|\beta_*\|_2^2 \right\},$$

*where* $\widetilde{\beta}$ *is the estimator obtained from solving problem* (7).

Theorem 3.4 shows that the upper bound of the prediction error mainly depends on the weight $\boldsymbol{\omega}$ and perturbation magnitude $\delta$. We apply the arguments in concentration inequalities and obtain the convergence rate in the following corollary.

**Corollary 3.5.** *Consider the group adversarial training problem* (7) *satisfying*

$$\frac{\delta}{\omega_l} = \frac{2}{\sqrt{\frac{2}{\pi} - \frac{1}{10}}} \sqrt{\frac{3p_l + 9\log L}{n}}, \forall l,$$

and $\Psi_l = \boldsymbol{X}_{G_l}^\top \boldsymbol{X}_{G_l}/n = I_{p_l \times p_l}$, where $\boldsymbol{X}_{G_l}$ denotes the $n \times p_l$ sub-matrix of $\boldsymbol{X}$ formed by the columns indexed by $G_l$. If $\beta_*$ is supported on a subset $G_J$ of $\{1, ..., p\}$ where $|J| \leq g$, and the design matrix $\boldsymbol{X}$ satisfies $\mathrm{GRE}(g, 3)$ with parameter $\kappa(g, 3)$, then we have that

$$\frac{1}{2n} \|\boldsymbol{X}(\widetilde{\beta} - \beta_*)\|_2^2 \leq 432 \frac{|G_J| + g \log L}{n} \max \left\{ \frac{9}{\kappa^2(s, 3)} \left( \frac{\|\boldsymbol{\epsilon}\|_1}{n} \right)^2, 164\|\beta_*\|_2^2 \right\}$$

holds with a probability greater than $1 - 2\exp(-C_1 n) - 2/L$, where $C_1$ is a positive constant.

**Remark 3.6.** *Note that we assume the gram matrix satisfies that $\boldsymbol{X}_{G_l}^\top \boldsymbol{X}_{G_l}/n = I_{p_l \times p_l}$. This is a standard assumption in the theoretical analysis in sparse high-dimensional linear regression, seeing [17, 19].*

Similar to the analytic investigations in Section 2, the convergence rate of the prediction error could be further simplified in the following corollary if the $\ell_2$-norm of the ground-truth parameter $\beta_*$ and error variance $\sigma^2$ are bounded.

**Corollary 3.7.** *Under the assumptions stated in Corollary 3.5, suppose there exists a finite positive constant $R$ such that*

$$2\sqrt{41}\|\beta_*\|_2 \leq R, \quad \sigma < \frac{1}{6}\kappa(g, 3)R,$$

*then we have that*

$$\frac{1}{2n}\|\boldsymbol{X}(\widetilde{\beta} - \beta_*)\|_2^2 \leq 432 \frac{|G_J| + g \log L}{n} R^2$$

*holds with a probability greater than $1 - 2/L - 2\exp(-C_1 n) - \exp(-n)$, where $C_1$ is a positive constant.*

**Remark 3.8.** *If we make the number of groups equal to $p$, i.e., each group only has one component, then we will have that $L = p, p_l = 1, |G_J| = g$. The resulting error bound is $g \log p/n$, where $g$ denotes the number of nonzero components of $\beta_*$. This order matches what is derived in Corollary 2.6.*

Corollary 3.7 indicates that the upper bound of the associated prediction error in group adversarial training under $(2, \infty)$-perturbation is of the order $(|G_J| + g \log L)/n$. Recall that $L$ is the number of prescribed groups for the $p$-dimensional variable, the ground-truth parameter $\beta_* \in \mathbb{R}^p$ is supported by a subset of the $L$ groups, and the subset is denoted by $J \subset \{1, ..., L\}$. The cardinality of the support subset $J$ is $g$. We also use $G_J \subset \{1, ..., p\}$ to denote all the indexes included in $J$.

It follows from Corollary 2.6 that the convergence rate of the prediction error for the classic adversarial training under $\ell_\infty$-perturbation is of the order $s \log p/n$, where $s$ denotes the cardinality of the support set of $\beta_*$. Then, we can conclude that if $|G_J|/s \ll \log p$ and $g \ll s$, the group adversarial training is superior to the classic adversarial training. In essence, if the sparsity pattern of $\beta_*$ has a good group structure, i.e., most of the nonzero components can be captured in $J$, then the group adversarial training procedure can provide an improved upper bound for the prediction error.

# 4  Numerical Experiments

In this section, we will run numerical experiments to observe the empirical performances of (group) adversarial training in high-dimensional linear regression.

We consider the following models to generate synthetic data: The response variable $Y$ is generated by the Gaussian linear model, as stated in Assumption 1.2. The standard deviation of the error $\boldsymbol{\epsilon}$ is chosen as $0.1$. In Model 1, the ground truth parameter $\beta_*$ is a 500-dimensional vector. The first four components are $[0.1, 0.2, 0.15, 0.25]$, the last four components are $[0.9, 0.95, 1, 1.05]$, and the other components are zero. In Model 2, the ground truth parameter $\beta_*$ is a 600-dimensional vector. The first three components of $\beta_*$ are $[0.4, 0.5, 0.6]$. The last three components of $\beta_*$ correspond to dummy variables generated from a four-level categorical factor. These dummy variable components are assigned values $[0.2, 0.3, 0.7]$. The other components are zero.

We run adversarial training under $\ell_\infty$-perturbation and group adversarial training under $(2, \infty)$-perturbation to give the estimations for the ground-truth parameter $\beta_*$, respectively. As suggested in Corollary 2.4 and Corollary 3.5, the perturbation magnitude is chosen in the order of $1/\sqrt{n}$ in the

adversarial training; the ratio of the perturbation magnitude and the perturbation weight is chosen in the order of $1/\sqrt{n}$ in the group adversarial training. For the constant, we selected $1$ for simplicity and experimental convenience. For the group adversarial training, we divide the parameter equally into 125 groups of size 4 for Model 1 and 200 groups of size 3 for Model 2. The sample sizes are chosen $\{50, 100, 150, 200, 250, 300, 350, 400\}$ for Model 1 and $\{50, 100, 150, 200, 250, 350, 450, 550\}$ for Model 2. In terms of computation, we apply the dual formulations, i.e., problem (2) and problem (8), and solve these convex optimization problems using the CVXPY toolbox [8]. Five random samples are generated at each sample size, and we run (group) adversarial training for each sample. The mean and standard error of the coefficient estimations and prediction errors are computed and recorded.

We first plot the coefficient estimation paths of adversarial training with error bars in Figure 1 and Figure 2. Both adversarial training and group adversarial training can shrink the parameter estimation, while group adversarial training performs a better shrinkage effect, In addition, the final values that the coefficients converge to are annotated in the figures. Given the ground-truth non-zero values $[0.1, 0.15, 0.2, 0.25, 0.9, 0.95, 1, 1.05]$ and $[0.4, 0.5, 0.6, 0.2, 0.3, 0.7]$, the final values of group adversarial training are closer to the ground-truth, indicating that the group adversarial training output more accurate estimations.

We also plot the curve of $\log_{10}(\text{prediction error})$ versus $\log_{10}(\text{sample size})$ with error bars in Figure 3 and Figure 4. We can observe that the slopes of two curves are approximately equal to $-1$, which is consistent with our theoretical analysis, where we have proved that the prediction error for high-dimensional (group) adversarial training is of the order $1/n$ in terms of the sample size $n$. Further, the curves and error bars of group adversarial training are below those of adversarial training, indicating the superiority of group adversarial training. This conclusion is also consistent with our theoretical analysis that if the model has a good group structure, group adversarial training has a lower order of prediction error.

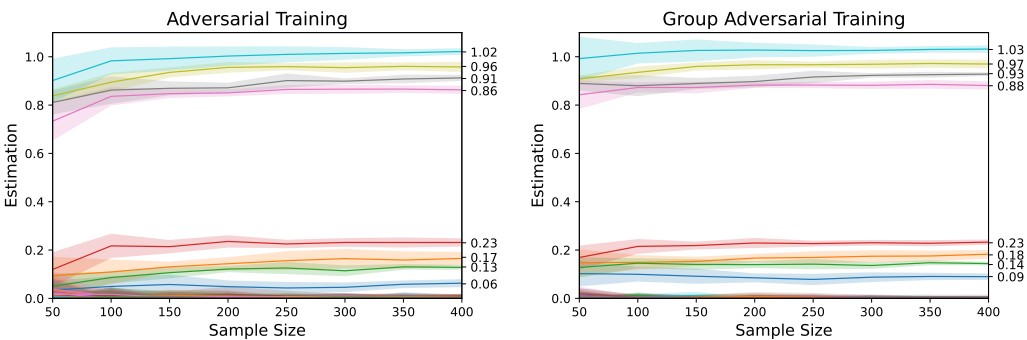

Figure 1: Coefficient Estimation Path in Model 1

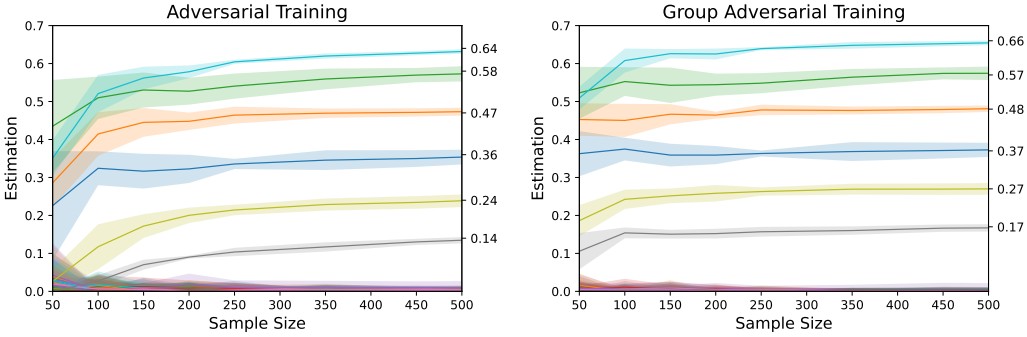

Figure 2: Coefficient Estimation Path in Model 2

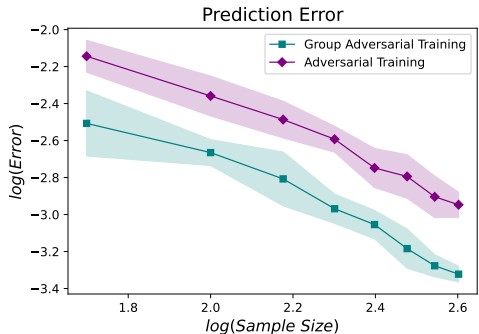
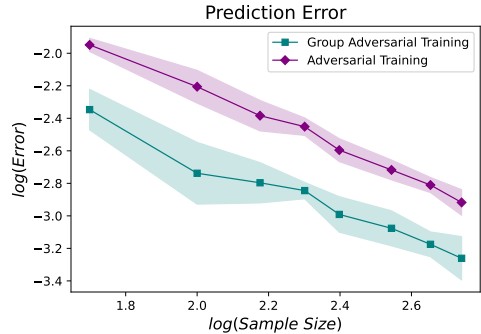

| Figure 3: Prediction Error in Model 1 | Figure 4: Prediction Error in Model 2 |

## 5  Discussions

This paper reveals the statistical optimality of adversarial training under $\ell_\infty$-perturbation in high dimensional linear regression and discusses potential improvements that can be achieved by group adversarial training. In the future, we may generalize the analysis in linear regression to broader statistical models, e.g., the generalized linear model and other parametric models. Also, since the prediction errors are investigated in this paper, we will consider analyzing estimation errors as our next step. More advanced analytical future work may use the primal-dual witness technique for the non-asymptotic variable-selection analysis.

## Acknowledgments and Disclosure of Funding

The authors would like to thank the chairs and anonymous reviewers for their careful comments, which helped enhance the presentation of the manuscript. The authors are partially sponsored by a subcontract of NSF grant 2229876, the A. Russell Chandler III Professorship at Georgia Institute of Technology, and NIH-sponsored Georgia Clinical & Translational Science Alliance.

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

# Appendix

## A  Proof of Theorem 2.3

We first give the upper bound of the $\|\widehat{\beta}\|_1$ in terms of $\|\beta_*\|_1$.

**Lemma A.1** (Upper Bound of $\|\widehat{\beta}\|_1$). *Under conditions stated in Theorem 2.3, we have that*

$$\|\widehat{\beta}\|_1 \le 9\|\beta_*\|_1.$$

*Proof.* The proof of this lemma follows a similar approach to the proof of Theorem 2 in [21]. It follows from the first-order condition of dual formulation (2) of the adversarial training problem that

$$0 = \frac{1}{n}\boldsymbol{X}^\top(\boldsymbol{X}\widehat{\beta} - \boldsymbol{Y}) + \delta^2\|\widehat{\beta}\|_1 w + \frac{\delta}{n}\|\boldsymbol{X}\widehat{\beta} - \boldsymbol{Y}\|_1 w + \frac{\delta}{n}\|\widehat{\beta}\|_1 \boldsymbol{X}^\top z, \tag{9}$$

where

$$z_i = \partial|\boldsymbol{X}_i^\top \widehat{\beta} - Y|, w_i = \partial|\widehat{\beta}|_i.$$

Then, we take the dot product of both sides of equation (9) with $\widehat{\beta} - \beta_*$ and could have the following:

$$
\frac{1}{n}\|\boldsymbol{X}(\widehat{\beta} - \beta_*)\|_2^2
$$
$$
= \frac{1}{n}(\boldsymbol{X}(\widehat{\beta} - \beta_*))^\top \boldsymbol{\epsilon} - \delta^2\|\widehat{\beta}\|_1 w^\top(\widehat{\beta} - \beta_*) - \frac{\delta}{n}\|\boldsymbol{X}\widehat{\beta} - \boldsymbol{Y}\|_1 w^\top(\widehat{\beta} - \beta_*) - \frac{\delta}{n}\|\widehat{\beta}\|_1(\boldsymbol{X}(\widehat{\beta} - \beta_*))^\top z,
$$
$$
= \frac{1}{n}(\boldsymbol{X}(\widehat{\beta} - \beta_*))^\top \boldsymbol{\epsilon} - \delta^2\|\widehat{\beta}\|_1^2 + \delta^2\|\widehat{\beta}\|_1 w^\top\beta_* - \frac{\delta}{n}\|\boldsymbol{X}\widehat{\beta} - \boldsymbol{Y}\|_1\|\widehat{\beta}\|_1 + \frac{\delta}{n}\|\boldsymbol{X}\widehat{\beta} - \boldsymbol{Y}\|_1 w^\top\beta_*
$$
$$
\quad - \frac{\delta}{n}\|\widehat{\beta}\|_1\|\boldsymbol{X}\widehat{\beta} - \boldsymbol{Y}\|_1 - \frac{\delta}{n}\|\widehat{\beta}\|_1 \boldsymbol{\epsilon}^\top z
$$
$$
\overset{(a)}{\le} \frac{\delta}{2n}\|\boldsymbol{\epsilon}\|_1\|\widehat{\beta} - \beta_*\|_1 - \delta^2\|\widehat{\beta}\|_1^2 + \delta^2\|\widehat{\beta}\|_1\|\beta_*\|_1 - \frac{\delta}{n}\|\boldsymbol{X}\widehat{\beta} - \boldsymbol{Y}\|_1\|\widehat{\beta}\|_1 + \frac{\delta}{n}\|\boldsymbol{X}\widehat{\beta} - \boldsymbol{Y}\|_1\|\beta_*\|_1
$$
$$
\quad - \frac{\delta}{n}\|\widehat{\beta}\|_1\|\boldsymbol{X}\widehat{\beta} - \boldsymbol{Y}\|_1 + \frac{\delta}{n}\|\widehat{\beta}\|_1\|\boldsymbol{\epsilon}\|_1
$$
$$
\overset{(b)}{\le} \frac{\delta}{n}\|\widehat{\beta}\|_1(\|\boldsymbol{\epsilon}\|_1 - 2\|\boldsymbol{X}\widehat{\beta} - \boldsymbol{Y}\|_1) + \frac{\delta}{n}\|\boldsymbol{X}\widehat{\beta} - \boldsymbol{Y}\|_1\|\beta_*\|_1 + \frac{\delta}{2n}\|\boldsymbol{\epsilon}\|_1(\|\widehat{\beta}\|_1 + \|\beta_*\|_1)
$$
$$
\quad - \delta^2\|\widehat{\beta}\|_1^2 + \delta^2\|\widehat{\beta}\|_1\|\beta_*\|_1
$$
$$
\overset{(c)}{\le} \frac{\delta}{n}\|\widehat{\beta}\|_1(\frac{5}{3}\|\boldsymbol{X}(\widehat{\beta} - \beta_*)\|_1 - \frac{2}{3}\|\boldsymbol{\epsilon}\|_1) + \frac{\delta}{n}\|\boldsymbol{X}\widehat{\beta} - \boldsymbol{Y}\|_1\|\beta_*\|_1 + \frac{\delta}{2n}\|\boldsymbol{\epsilon}\|_1(\|\widehat{\beta}\|_1 + \|\beta_*\|_1)
$$
$$
\quad - \delta^2\|\widehat{\beta}\|_1^2 + \delta^2\|\widehat{\beta}\|_1\|\beta_*\|_1
$$
$$
\overset{(d)}{\le} \frac{\delta}{\sqrt{n}}\|\boldsymbol{X}(\widehat{\beta} - \beta_*)\|_2(\frac{5}{3}\|\widehat{\beta}\|_1 + \|\beta_*\|_1) + \frac{\delta}{n}\|\boldsymbol{\epsilon}\|_1(-\frac{1}{6}\|\widehat{\beta}\|_1 + \frac{3}{2}\|\beta_*\|_1) - \delta^2\|\widehat{\beta}\|_1^2 + \delta^2\|\widehat{\beta}\|_1\|\beta_*\|_1,
$$
$$\tag{10}$$

where (a) comes from the Hölder's inequality, i.e.,

$$(\boldsymbol{X}(\widehat{\beta} - \beta_*))^\top \boldsymbol{\epsilon} \le \|\widehat{\beta} - \beta_*\|_1\|\boldsymbol{X}^\top \boldsymbol{\epsilon}\|_\infty, \ w^\top\beta_* \le \|w\|_\infty\|\beta_*\|_1 = \|\beta_*\|_1, \ -\boldsymbol{\epsilon}^\top z \le \|\boldsymbol{\epsilon}\|_1\|z\|_\infty = \|\boldsymbol{\epsilon}\|_1,$$

and the condition $2\|\boldsymbol{X}^\top \boldsymbol{\epsilon}\|_\infty/\|\boldsymbol{\epsilon}\|_1 \le \delta$, (b) comes from $\|\widehat{\beta} - \beta_*\|_1 \le \|\widehat{\beta}\|_1 + \|\beta_*\|_1$, (c) comes from the relationship that

$$2\|\boldsymbol{X}\widehat{\beta} - \boldsymbol{Y}\|_1 \ge \frac{5}{3}\|\boldsymbol{X}\widehat{\beta} - \boldsymbol{Y}\|_1 \ge -\frac{5}{3}\|\boldsymbol{X}(\widehat{\beta} - \beta_*)\|_1 + \frac{5}{3}\|\boldsymbol{\epsilon}\|_1,$$

and (d) comes from the inequality that $\|\boldsymbol{X}(\widehat{\beta} - \beta_*)\|_1 \le \sqrt{n}\|\boldsymbol{X}(\widehat{\beta} - \beta_*)\|_2$ and $\|\boldsymbol{X}\widehat{\beta} - \boldsymbol{Y}\|_1 \le \|\boldsymbol{X}(\widehat{\beta} - \beta_*)\|_1 + \|\boldsymbol{\epsilon}\|_1$.

One may observe that (10) is a second-order inequality of variable $\|\boldsymbol{X}(\widehat{\beta} - \beta_*)\|_2/\sqrt{n}$, then the associate discriminant should be equal to or larger than 0, resulting in

$$\frac{1}{9}\delta^2(-11\|\widehat{\beta}\|_1^2 + 66\|\beta_*\|_1\|\widehat{\beta}\|_1 + 9\|\beta_*\|_1^2) + \frac{\delta}{3n}\|\boldsymbol{\epsilon}\|_1(-2\|\widehat{\beta}\|_1 + 18\|\beta_*\|_1) \ge 0,$$

from which we could conclude that at least one of two terms should be equal or larger than 0, implying

$$\|\widehat{\beta}\|_1 \le 9\|\beta_*\|_1.$$

$\square$

**Then, we proceed to prove Theorem 2.3.**

*Proof.* We write the objective function in (2) in the matrix norm and then have that

$$\frac{1}{2n}\|\boldsymbol{Y}-\boldsymbol{X}\widehat{\beta}\|_2^2+\frac{1}{2}\delta^2\|\widehat{\beta}\|_1^2+\frac{\delta}{n}\|\boldsymbol{Y}-\boldsymbol{X}\widehat{\beta}\|_1\|\widehat{\beta}\|_1 \le \frac{1}{2n}\|\boldsymbol{Y}-\boldsymbol{X}\beta_*\|_2^2+\frac{1}{2}\delta^2\|\beta_*\|_1^2+\frac{\delta}{n}\|\boldsymbol{Y}-\boldsymbol{X}\beta_*\|_1\|\beta_*\|_1.$$
(11)

It follows from $\boldsymbol{Y} = \boldsymbol{X}\beta_* + \boldsymbol{\epsilon}$, i.e., Assumption 1.2, that

$$\|\boldsymbol{Y} - \boldsymbol{X}\beta_*\|_2^2 = \|\boldsymbol{\epsilon}\|_2^2,$$

$$\|\boldsymbol{Y} - \boldsymbol{X}\widehat{\beta}\|_2^2 = \|\boldsymbol{X}(\widehat{\beta} - \beta_*) - \boldsymbol{\epsilon}\|_2^2 = \|\boldsymbol{X}(\widehat{\beta} - \beta_*)\|_2^2 + \|\boldsymbol{\epsilon}\|_2^2 - 2\boldsymbol{\epsilon}^\top \boldsymbol{X}(\widehat{\beta} - \beta_*),$$

$$\|\boldsymbol{Y} - \boldsymbol{X}\widehat{\beta}\|_1 = \|\boldsymbol{X}(\widehat{\beta} - \beta_*) - \boldsymbol{\epsilon}\|_1 \ge \|\boldsymbol{\epsilon}\|_1 - \|\boldsymbol{X}(\widehat{\beta} - \beta_*)\|_1.$$

In this way, the inequality (11) could be reformulated as

$$\frac{1}{2n}\|\boldsymbol{X}(\widehat{\beta} - \beta_*)\|_2^2$$

$$\le \frac{1}{n}\boldsymbol{\epsilon}^\top \boldsymbol{X}(\widehat{\beta} - \beta_*) + \frac{\delta}{n}\|\boldsymbol{X}(\widehat{\beta} - \beta_*)\|_1\|\widehat{\beta}\|_1 + \frac{\delta}{n}\|\boldsymbol{\epsilon}\|_1\left(\|\beta_*\|_1 - \|\widehat{\beta}\|_1\right) + \frac{1}{2}\delta^2\|\beta_*\|_1^2 - \frac{1}{2}\delta^2\|\widehat{\beta}\|_1^2$$

$$\overset{(a)}{\le} \frac{1}{n}\|\boldsymbol{X}^\top \boldsymbol{\epsilon}\|_\infty\|\widehat{\beta} - \beta_*\|_1 + \frac{\delta}{n}\|\boldsymbol{X}(\widehat{\beta} - \beta_*)\|_1\|\widehat{\beta}\|_1 + \frac{\delta}{n}\|\boldsymbol{\epsilon}\|_1\left(\|\beta_*\|_1 - \|\widehat{\beta}\|_1\right) + \frac{1}{2}\delta^2\|\beta_*\|_1^2 - \frac{1}{2}\delta^2\|\widehat{\beta}\|_1^2$$

$$\overset{(b)}{\le} \frac{\delta}{2n}\|\boldsymbol{\epsilon}\|_1\|\widehat{\beta} - \beta_*\|_1 + \frac{\delta}{n}\|\boldsymbol{X}(\widehat{\beta} - \beta_*)\|_1\|\widehat{\beta}\|_1 + \frac{\delta}{n}\|\boldsymbol{\epsilon}\|_1\left(\|\beta_*\|_1 - \|\widehat{\beta}\|_1\right) + \frac{1}{2}\delta^2\|\beta_*\|_1^2 - \frac{1}{2}\delta^2\|\widehat{\beta}\|_1^2$$

$$\overset{(c)}{\le} \frac{\delta}{2n}\|\boldsymbol{\epsilon}\|_1\|\widehat{\beta} - \beta_*\|_1 + \frac{\delta}{\sqrt{n}}\|\boldsymbol{X}(\widehat{\beta} - \beta_*)\|_2\|\widehat{\beta}\|_1 + \frac{\delta}{n}\|\boldsymbol{\epsilon}\|_1\left(\|\beta_*\|_1 - \|\widehat{\beta}\|_1\right) + \frac{1}{2}\delta^2\|\beta_*\|_1^2 - \frac{1}{2}\delta^2\|\widehat{\beta}\|_1^2,$$
(12)

where (a) comes from the Hölder's inequality, (b) comes from the condition $2\|\boldsymbol{X}^\top \boldsymbol{\epsilon}\|_\infty/\|\boldsymbol{\epsilon}\|_1 \le \delta$, (c) comes from $\|\boldsymbol{X}(\widehat{\beta} - \beta_*)\|_1 \le \sqrt{n}\|\boldsymbol{X}(\widehat{\beta} - \beta_*)\|_2$.

Then, we begin to give the upper bound of the prediction error. Two cases should be discussed.

**First Case:**
$$\|\widehat{\beta} - \beta_*\|_1 + 2\|\beta_*\|_1 - 2\|\widehat{\beta}\|_1 \le 0.$$
(13)

In this case, we have that

$$\|\beta_*\|_1 - \|\widehat{\beta}\|_1 = \|\widehat{\beta}\|_1 - \|\beta_*\|_1 + 2\|\beta_*\|_1 - 2\|\widehat{\beta}\|_1 \le \|\widehat{\beta} - \beta_*\|_1 + 2\|\beta_*\|_1 - 2\|\widehat{\beta}\|_1 \le 0. \quad (14)$$

Then, it follows from (12) that

$$\frac{1}{2n}\|\boldsymbol{X}(\widehat{\beta} - \beta_*)\|_2^2$$

$$\le \frac{\delta}{2n}\|\boldsymbol{\epsilon}\|_1\|\widehat{\beta} - \beta_*\|_1 + \frac{\delta}{\sqrt{n}}\|\boldsymbol{X}(\widehat{\beta} - \beta_*)\|_2\|\widehat{\beta}\|_1 + \frac{\delta}{n}\|\boldsymbol{\epsilon}\|_1\left(\|\beta_*\|_1 - \|\widehat{\beta}\|_1\right) + \frac{1}{2}\delta^2\|\beta_*\|_1^2 - \frac{1}{2}\delta^2\|\widehat{\beta}\|_1^2$$

$$= \frac{\delta}{2n}\|\boldsymbol{\epsilon}\|_1\left(\|\widehat{\beta} - \beta_*\|_1 + 2\|\beta_*\|_1 - 2\|\widehat{\beta}\|_1\right) + \frac{\delta}{\sqrt{n}}\|\boldsymbol{X}(\widehat{\beta} - \beta_*)\|_2\|\widehat{\beta}\|_1 + \frac{1}{2}\delta^2\|\beta_*\|_1^2 - \frac{1}{2}\delta^2\|\widehat{\beta}\|_1^2$$

$$\le \frac{\delta}{\sqrt{n}}\|\boldsymbol{X}(\widehat{\beta} - \beta_*)\|_2\|\widehat{\beta}\|_1,$$

where the last inequality comes from (13) and (14).

Lemma A.1 indicates that

$$\frac{1}{\sqrt{2n}}\|\boldsymbol{X}(\widehat{\beta} - \beta_*)\|_2 \le \sqrt{2}\delta\|\widehat{\beta}\|_1 \le 9\sqrt{2}\delta\|\beta_*\|_1,$$

which is equivalent to

$$\frac{1}{2n}\|\boldsymbol{X}(\widehat{\beta}-\beta_*)\|_2^2 \leq 162\delta^2\|\beta_*\|_1^2 \leq 162\delta^2|S|\|\beta_*\|_2^2 \leq 162\delta^2 s\|\beta_*\|_2^2. \tag{15}$$

**Second Case:**
$$\|\widehat{\beta}-\beta_*\|_1 + 2\|\beta_*\|_1 - 2\|\widehat{\beta}\|_1 \geq 0. \tag{16}$$

If we let $\widehat{v} = \widehat{\beta} - \beta_*$, then we have that

$$\|\beta_* - \widehat{\beta}\|_1 = \|\widehat{v}\|_1 = \|\widehat{v}_S\|_1 + \|\widehat{v}_{S^c}\|_1,$$

$$\|\beta_*\|_1 - \|\widehat{\beta}\|_1 = \|\beta_{*S}\|_1 - (\|\beta_{*S} + \widehat{v}_S\|_1 + \|\widehat{v}_{S^c}\|_1) \leq \|\widehat{v}_S\|_1 - \|\widehat{v}_{S^c}\|_1.$$

The inequality (16) indicates that

$$3\|\widehat{v}_S\|_1 - \|\widehat{v}_{S^c}\|_1 \geq \|\widehat{\beta}-\beta_*\|_1 + 2\|\beta_*\|_1 - 2\|\widehat{\beta}\|_1 \geq 0, \tag{17}$$

implying
$$\|\widehat{v}_{S^c}\|_1 \leq 3\|\widehat{v}_S\|_1.$$

In this way, the $\mathrm{RE}(s,3)$ condition can be applied.

In addition, it follows the inequality (12) that

$$\frac{1}{2n}\|\boldsymbol{X}(\widehat{\beta}-\beta_*)\|_2^2$$

$$\leq \frac{\delta}{2n}\|\epsilon\|_1\|\widehat{\beta}-\beta_*\|_1 + \frac{\delta}{\sqrt{n}}\|\boldsymbol{X}(\widehat{\beta}-\beta_*)\|_2\|\widehat{\beta}\|_1 + \frac{\delta}{n}\|\epsilon\|_1\left(\|\beta_*\|_1 - \|\widehat{\beta}\|_1\right) + \frac{1}{2}\delta^2\|\beta_*\|_1^2 - \frac{1}{2}\delta^2\|\widehat{\beta}\|_1^2$$

$$\overset{(a)}{\leq} \frac{\delta}{2n}\|\epsilon\|_1\|\widehat{\beta}-\beta_*\|_1 + \frac{1}{4n}\|\boldsymbol{X}(\widehat{\beta}-\beta_*)\|_2^2 + \delta^2\|\widehat{\beta}\|_1^2 + \frac{\delta}{n}\|\epsilon\|_1\left(\|\beta_*\|_1 - \|\widehat{\beta}\|_1\right) + \frac{1}{2}\delta^2\|\beta_*\|_1^2 - \frac{1}{2}\delta^2\|\widehat{\beta}\|_1^2$$

$$= \frac{\delta}{2n}\|\epsilon\|_1\left(\|\widehat{\beta}-\beta_*\|_1 + 2\|\beta_*\|_1 - 2\|\widehat{\beta}\|_1\right) + \frac{1}{4n}\|\boldsymbol{X}(\widehat{\beta}-\beta_*)\|_2^2 + \frac{1}{2}\delta^2\|\beta_*\|_1^2 + \frac{1}{2}\delta^2\|\widehat{\beta}\|_1^2$$

$$\overset{(b)}{\leq} \frac{3\delta}{2n}\|\epsilon\|_1\|\widehat{v}_S\|_1 + \frac{1}{4n}\|\boldsymbol{X}(\widehat{\beta}-\beta_*)\|_2^2 + 41\delta^2\|\beta_*\|_1^2,$$

where (a) comes from the inequality

$$\frac{1}{4n}\|\boldsymbol{X}(\widehat{\beta}-\beta_*)\|_2^2 + \delta^2\|\widehat{\beta}\|_1^2 \geq \frac{\delta}{\sqrt{n}}\|\boldsymbol{X}(\widehat{\beta}-\beta_*)\|_2\|\widehat{\beta}\|_1,$$

(b) comes from (17) and Lemma A.1.

Further, we have that

$$\frac{1}{4n}\|\boldsymbol{X}(\widehat{\beta}-\beta_*)\|_2^2 \leq \frac{3\delta}{2n}\sqrt{|S|}\|\epsilon\|_1\|\widehat{v}\|_2 + 41\delta^2|S|\|\beta_*\|_2^2$$

$$\leq \frac{3\delta}{2\sqrt{n}}\frac{\sqrt{s}}{\gamma(s,3)}\frac{\|\epsilon\|_1}{n}\|\boldsymbol{X}(\widehat{\beta}-\beta_*)\|_2 + 41\delta^2 s\|\beta_*\|_2^2, \tag{18}$$

where the first inequality comes from $\|\widehat{v}_S\|_1 \leq \sqrt{|S|}\|\widehat{v}\|_2$ and $\|\beta_*\|_1 \leq \sqrt{|S|}\|\beta_*\|_2$, and the last inequality comes from the $\mathrm{RE}(s,3)$ condition and $|S| \leq s$.

Then, if we solve the inequality (18), we could have that

$$\frac{1}{\sqrt{n}}\|\boldsymbol{X}(\widehat{\beta}-\beta_*)\|_2 \leq \frac{\delta\sqrt{s}}{\gamma(s,3)}\left(3\frac{\|\epsilon\|_1}{n} + \sqrt{9\left(\frac{\|\epsilon\|_1}{n}\right)^2 + 164\gamma^2(s,3)\|\beta_*\|_2^2}\right),$$

which is equivalent to

$$\frac{1}{\sqrt{n}}\|\boldsymbol{X}(\widehat{\beta}-\beta_*)\|_2 \leq (1+\sqrt{2})\frac{\delta\sqrt{s}}{\gamma(s,3)}\max\left\{3\frac{\|\epsilon\|_1}{n}, \sqrt{164}\gamma(s,3)\|\beta_*\|_2\right\},$$

indicating that

$$\frac{1}{2n}\|\boldsymbol{X}(\widehat{\beta} - \beta_*)\|_2^2 \leq 3\frac{\delta^2 s}{\gamma^2(s,3)} \max\left\{9\left(\frac{\|\boldsymbol{\epsilon}\|_1}{n}\right)^2, 164\gamma^2(s,3)\|\beta_*\|_2^2\right\}. \tag{19}$$

Combining (15) and (19), we have that

$$\frac{1}{2n}\|\boldsymbol{X}(\widehat{\beta} - \beta_*)\|_2^2 \leq 3\delta^2 s \max\left\{\frac{9}{\gamma^2(s,3)}\left(\frac{\|\boldsymbol{\epsilon}\|_1}{n}\right)^2, 164\|\beta_*\|_2^2\right\}.$$

$\square$

## B  Proof of Corollary 2.4

*Proof.* To give the high probability result, we should analyze the bound of the term $\|\boldsymbol{X}^\top\boldsymbol{\epsilon}\|_\infty/n$ and $\|\boldsymbol{\epsilon}\|_1/n$. The associated arguments are discussed in the following two parts, respectively. (Note that we assume $\boldsymbol{\epsilon}$ has i.i.d. Gaussian entries. However, our high-probability result can be extended to sub-Gaussian cases, as sub-Gaussian variables share similar tail decay behavior.)

**Part I**: We focus on the tail bound of $\|\boldsymbol{X}^\top\boldsymbol{\epsilon}\|_\infty/n$. Since the design matrix $\boldsymbol{X}$ is normalized, the random variable $\boldsymbol{x}_j^\top\boldsymbol{\epsilon}/n$ is stochastically dominated by $\mathcal{N}(0, \sigma^2/n)$. As shown in Theorem 11.1 in [11], it follows from the Gaussian tail bound and the union bound that

$$\mathbb{P}\left(\frac{\|\boldsymbol{X}^\top\boldsymbol{\epsilon}\|_\infty}{n} \geq t\right) \leq 2\exp\left(-\frac{nt^2}{2\sigma^2} + \log p\right).$$

In this way, with a probability greater than $1 - 2/p$, the following holds

$$\frac{\|\boldsymbol{X}^\top\boldsymbol{\epsilon}\|_\infty}{n} \leq 2\sigma\sqrt{\frac{\log p}{n}}.$$

**Part II**: We focus on the concentration inequality of $\|\boldsymbol{\epsilon}\|_1/n$. It follows from general Hoeffding's inequality [26] that

$$\mathbb{P}\left(\left|\|\boldsymbol{\epsilon}\|_1 - n\sigma\sqrt{\frac{2}{\pi}}\right| \geq t\right) \leq 2\exp\left(-C_2\frac{t^2}{n\sigma^2}\right),$$

indicating

$$\mathbb{P}\left(\left|\frac{1}{n}\|\boldsymbol{\epsilon}\|_1 - \sigma\sqrt{\frac{2}{\pi}}\right| \geq t\right) \leq 2\exp\left(-C_2\frac{nt^2}{\sigma^2}\right),$$

where $C_2$ is some positive constant. By choosing $t = \sigma/10$, we have that

$$\mathbb{P}\left(\sigma\left(\sqrt{\frac{2}{\pi}} - \frac{1}{10}\right) \leq \frac{1}{n}\|\boldsymbol{\epsilon}\|_1\right) \geq 1 - 2\exp\left(-C_1 n\right),$$

where $C_1$ is some positive constant. This is to say, with a probability greater than $1 - 2\exp\left(-C_1 n\right)$, we have that

$$\sigma\left(\sqrt{\frac{2}{\pi}} - \frac{1}{10}\right) \leq \frac{1}{n}\|\boldsymbol{\epsilon}\|_1.$$

Suppose we let

$$\delta = \frac{4}{\sqrt{\frac{2}{\pi}} - \frac{1}{10}}\sqrt{\frac{\log p}{n}},$$

with a probability greater than $1 - 2\exp\left(-C_1 n\right) - 2/p$, we have that

$$\frac{2\|\boldsymbol{X}^\top\boldsymbol{\epsilon}\|_\infty}{\|\boldsymbol{\epsilon}\|_1} \leq \delta.$$

It follows from Theorem 2.3 that

$$\frac{1}{2n^2}\|\boldsymbol{X}(\widehat{\beta} - \beta_*)\|_2^2 \leq 192\frac{s\log p}{n}\max\left\{\frac{9}{\gamma^2(s,3)}\left(\frac{\|\boldsymbol{\epsilon}\|_1}{n}\right)^2, 164\|\beta_*\|_2^2\right\}$$

holds with a probability greater $1 - 2\exp\left(-C_1 n\right) - 2/p$. $\square$

## C   Proof of Corollary 2.6

*Proof.* We focus on the tail bound of $\|\boldsymbol{\epsilon}\|_1/n$. We apply the Chernoff bound to give the tail bound of $\|\boldsymbol{\epsilon}\|_1$, and we have that

$$\mathbb{P}\left(\|\boldsymbol{\epsilon}\|_1 \geq t\right) \leq \inf_{s>0} M(s)\exp(-ts),$$

where $M(s)$ is the moment-generating function of $\|\boldsymbol{\epsilon}\|_1$. We also could obtain that

$$M_i(s) = \mathbb{E}[\exp(s|\boldsymbol{\epsilon}_i|)] \leq 2\mathbb{E}[\exp(s\boldsymbol{\epsilon}_i)] = 2\exp\left(\frac{\sigma^2 s^2}{2}\right),$$

indicating

$$M(s) \leq 2^n \exp\left(\frac{n\sigma^2 s^2}{2}\right).$$

Then, we have that

$$\mathbb{P}\left(\|\boldsymbol{\epsilon}\|_1 \geq t\right) \leq \inf_{s>0} 2^n \exp\left(\frac{n\sigma^2 s^2}{2} - ts\right),$$

indicating

$$\mathbb{P}\left(\frac{1}{n}\|\boldsymbol{\epsilon}\|_1 \geq t\right) \leq \exp\left(-\frac{nt^2}{2\sigma^2} + n\log 2\right).$$

In this way, with a probability greater $1 - \exp(-n)$, the following holds:

$$\frac{1}{n}\|\boldsymbol{\epsilon}\|_1 \leq \sqrt{2\log 2 + 2}\sigma \leq 2\sigma.$$

Since we have that $\frac{1}{6}\gamma(s,3)R \geq \sigma$,

$$\frac{9}{\gamma^2(s,3)}\left(\frac{\|\boldsymbol{\epsilon}\|_1}{n}\right)^2 \leq R^2,$$

holds with a probability greater $1 - \exp(-n)$. Due to Corollary 2.4, we have the following

$$\frac{1}{2n^2}\|\boldsymbol{X}(\widehat{\beta} - \beta_*)\|_2^2 \leq 192\frac{s\log p}{n}R^2$$

holds with a probability greater $1 - 2/p - 2\exp(-C_1 n) - \exp(-n)$.   $\square$

## D   Proof of Proposition 3.2

It follows from Proposition 1 in [21] that

$$\widetilde{R}(\delta) = \min_{\beta} \frac{1}{n}\sum_{i=1}^{n}\left(|(\boldsymbol{X}_i + \Delta)^\top \beta - Y_i| + \delta\|\beta_{\boldsymbol{\omega}^{-1}}\|_*\right)^2,$$

where $\|\cdot\|_*$ denotes the dual norm of $(2,\infty)$-norm of $\beta_{\boldsymbol{\omega}}$. Due to Proposition 3.1, we conclude that (8) holds.

## E   Proof of Theorem 3.4

We first give the upper bound of the $\|\widetilde{\beta}_{\boldsymbol{\omega}^{-1}}\|_{2,1}$ in terms of $\|\beta_{*\boldsymbol{\omega}^{-1}}\|_{2,1}$.

**Lemma E.1** (Upper Bound of $\|\widetilde{\beta}_{\boldsymbol{\omega}^{-1}}\|_{2,1}$). *Under conditions stated in Theorem 3.4, we have that*

$$\|\widetilde{\beta}_{\boldsymbol{\omega}^{-1}}\|_{2,1} \leq 9\|\beta_{*\boldsymbol{\omega}^{-1}}\|_{2,1}.$$

*Proof.* The proof of this lemma follows a similar approach to the proof of Lemma A.1. It follows from the first-order condition of the dual formulation (8) of the group adversarial training problem that

$$\mathbf{0} = \frac{1}{n}\boldsymbol{X}^\top(\boldsymbol{X}\widetilde{\beta} - \boldsymbol{Y}) + \delta^2\|\widetilde{\beta}_{\boldsymbol{\omega}^{-1}}\|_{2,1}t + \frac{\delta}{n}\|\boldsymbol{X}\widetilde{\beta} - \boldsymbol{Y}\|_1 t + \frac{\delta}{n}\|\widetilde{\beta}_{\boldsymbol{\omega}^{-1}}\|_{2,1}\boldsymbol{X}^\top z, \qquad (20)$$

where

$$z_i = \partial |X_i^\top \widetilde{\beta} - Y|,$$

$$t = \partial \|\widetilde{\beta}_{\boldsymbol{\omega}^{-1}}\|_{2,1}, \quad t^l = \frac{1}{\omega_l} \frac{\widetilde{\beta}^l}{\|\widetilde{\beta}^l\|_2}.$$

Notice we have that

$$t^\top \widetilde{\beta} = \sum_{l=1}^L \frac{1}{\omega_l} \frac{(\widetilde{\beta}^l)^\top \widetilde{\beta}^l}{\|\widetilde{\beta}^l\|_2} = \sum_{l=1}^L \frac{1}{\omega_l} \|\widetilde{\beta}^l\|_2 = \|\widetilde{\beta}_{\boldsymbol{\omega}^{-1}}\|_{2,1}, \tag{21}$$

and it follows from the Hölder's inequality that

$$t^\top \beta_* = \sum_{l=1}^L \frac{1}{\omega_l} \frac{(\widetilde{\beta}^l)^\top \beta_*^l}{\|\widetilde{\beta}^l\|_2} \leq \sum_{l=1}^L \frac{1}{\omega_l} \|\beta_*^l\|_2 = \|\beta_{*\boldsymbol{\omega}^{-1}}\|_{2,1}. \tag{22}$$

Also, we have the following from the Hölder's inequality:

$$(\boldsymbol{X}(\widetilde{\beta} - \beta_*))^\top \boldsymbol{\epsilon} \leq \sum_{l=1}^L \|(\boldsymbol{X}^\top \boldsymbol{\epsilon})^l\|_2 \|\widetilde{\beta}^l - \beta_*^l\|_2 \leq \frac{1}{2}\delta \|\boldsymbol{\epsilon}\|_1 \sum_{l=1}^L \frac{1}{\omega_l} \|\widetilde{\beta}^l - \beta_*^l\|_2 = \frac{1}{2}\delta \|\boldsymbol{\epsilon}\|_1 \|(\widetilde{\beta}^l - \beta_*^l)_{\boldsymbol{\omega}^{-1}}\|_{2,1}, \tag{23}$$

where the second inequality comes from the condition $2\|\left(\boldsymbol{X}^\top \boldsymbol{\epsilon}\right)^l\|_2 / \|\boldsymbol{\epsilon}\|_1 \leq \delta/\omega_l$.

Then, we take the dot product of both sides of equation (20) with $\widetilde{\beta} - \beta_*$ and could have the following:

$$\frac{1}{n}\|\boldsymbol{X}(\widetilde{\beta} - \beta_*)\|_2^2$$

$$= \frac{1}{n}(\boldsymbol{X}(\widetilde{\beta} - \beta_*))^\top \boldsymbol{\epsilon} - \delta^2 \|\widetilde{\beta}_{\boldsymbol{\omega}^{-1}}\|_{2,1} t^\top (\widetilde{\beta} - \beta_*) - \frac{\delta}{n}\|\boldsymbol{X}\widetilde{\beta} - \boldsymbol{Y}\|_1 t^\top (\widetilde{\beta} - \beta_*)$$

$$\quad - \frac{\delta}{n}\|\widetilde{\beta}_{\boldsymbol{\omega}^{-1}}\|_{2,1}(\boldsymbol{X}(\widetilde{\beta} - \beta_*))^\top z,$$

$$\overset{(a)}{=} \frac{1}{n}(\boldsymbol{X}(\widetilde{\beta} - \beta_*))^\top \boldsymbol{\epsilon} - \delta^2 \|\widetilde{\beta}_{\boldsymbol{\omega}^{-1}}\|_{2,1}^2 + \delta^2 \|\widetilde{\beta}_{\boldsymbol{\omega}^{-1}}\|_{2,1} t^\top \beta_* - \frac{\delta}{n}\|\boldsymbol{X}\widetilde{\beta} - \boldsymbol{Y}\|\|\widetilde{\beta}_{\boldsymbol{\omega}^{-1}}\|_{2,1}$$

$$\quad + \frac{\delta}{n}\|\boldsymbol{X}\widetilde{\beta} - \boldsymbol{Y}\|_1 t^\top \beta_* - \frac{\delta}{n}\|\widetilde{\beta}_{\boldsymbol{\omega}^{-1}}\|_{2,1}\|\boldsymbol{X}\widetilde{\beta} - \boldsymbol{Y}\|_1 + \frac{\delta}{n}\|\widetilde{\beta}_{\boldsymbol{\omega}^{-1}}\|_{2,1}\boldsymbol{\epsilon}^\top z$$

$$\overset{(b)}{\leq} \frac{\delta}{2n}\|\boldsymbol{\epsilon}\|_1 \|(\widetilde{\beta}^l - \beta_*^l)_{\boldsymbol{\omega}^{-1}}\|_{2,1} - \delta^2 \|\widetilde{\beta}_{\boldsymbol{\omega}^{-1}}\|_{2,1}^2 + \delta^2 \|\widetilde{\beta}_{\boldsymbol{\omega}^{-1}}\|_{2,1}\|\beta_{*\boldsymbol{\omega}^{-1}}\|_{2,1} - \frac{\delta}{n}\|\boldsymbol{X}\widetilde{\beta} - \boldsymbol{Y}\|_1 \|\widetilde{\beta}_{\boldsymbol{\omega}^{-1}}\|_{2,1}$$

$$\quad + \frac{\delta}{n}\|\boldsymbol{X}\widetilde{\beta} - \boldsymbol{Y}\|_1 \|\beta_{*\boldsymbol{\omega}^{-1}}\|_{2,1} - \frac{\delta}{n}\|\widetilde{\beta}_{\boldsymbol{\omega}^{-1}}\|_{2,1}\|\boldsymbol{X}\widetilde{\beta} - \boldsymbol{Y}\|_1 + \frac{\delta}{n}\|\widetilde{\beta}_{\boldsymbol{\omega}^{-1}}\|_{2,1}\boldsymbol{\epsilon}^\top z$$

$$\overset{(c)}{\leq} \frac{\delta}{n}\|\widetilde{\beta}_{\boldsymbol{\omega}^{-1}}\|_{2,1}(\|\boldsymbol{\epsilon}\|_1 - 2\|\boldsymbol{X}\widetilde{\beta} - \boldsymbol{Y}\|_1) + \frac{\delta}{n}\|\boldsymbol{X}\widetilde{\beta} - \boldsymbol{Y}\|_1 \|\beta_{*\boldsymbol{\omega}^{-1}}\|_{2,1}$$

$$\quad + \frac{\delta}{2n}\|\boldsymbol{\epsilon}\|_1(\|\widetilde{\beta}_{\boldsymbol{\omega}^{-1}}\|_{2,1} + \|\beta_{*\boldsymbol{\omega}^{-1}}\|_{2,1}) - \delta^2\|\widetilde{\beta}_{\boldsymbol{\omega}^{-1}}\|_{2,1}^2 + \delta^2\|\widetilde{\beta}_{\boldsymbol{\omega}^{-1}}\|_{2,1}\|\beta_{*\boldsymbol{\omega}^{-1}}\|_{2,1}$$

$$\overset{(d)}{\leq} \frac{\delta}{n}\|\widetilde{\beta}_{\boldsymbol{\omega}^{-1}}\|_{2,1}(\frac{5}{3}\|\boldsymbol{X}(\widetilde{\beta} - \beta_*)\|_1 - \frac{2}{3}\|\boldsymbol{\epsilon}\|_1) + \frac{\delta}{n}\|\boldsymbol{X}\widetilde{\beta} - \boldsymbol{Y}\|_1 \|\beta_{*\boldsymbol{\omega}^{-1}}\|_{2,1}$$

$$\quad + \frac{\delta}{2n}\|\boldsymbol{\epsilon}\|_1(\|\widetilde{\beta}_{\boldsymbol{\omega}^{-1}}\|_{2,1} + \|\beta_{*\boldsymbol{\omega}^{-1}}\|_{2,1}) - \delta^2\|\widetilde{\beta}_{\boldsymbol{\omega}^{-1}}\|_{2,1}^2 + \delta^2\|\widetilde{\beta}_{\boldsymbol{\omega}^{-1}}\|_{2,1}\|\beta_{*\boldsymbol{\omega}^{-1}}\|_{2,1}$$

$$\overset{(e)}{\leq} \frac{\delta}{\sqrt{n}}\|\boldsymbol{X}(\widetilde{\beta} - \beta_*)\|_2(\frac{5}{3}\|\widetilde{\beta}_{\boldsymbol{\omega}^{-1}}\|_{2,1} + \|\beta_{*\boldsymbol{\omega}^{-1}}\|_{2,1}) + \frac{\delta}{n}\|\boldsymbol{\epsilon}\|_1(-\frac{1}{6}\|\widetilde{\beta}_{\boldsymbol{\omega}^{-1}}\|_{2,1} + \frac{3}{2}\|\beta_{*\boldsymbol{\omega}^{-1}}\|_{2,1})$$

$$\quad - \delta^2\|\widetilde{\beta}_{\boldsymbol{\omega}^{-1}}\|_{2,1}^2 + \delta^2\|\widetilde{\beta}_{\boldsymbol{\omega}^{-1}}\|_{2,1}\|\beta_{*\boldsymbol{\omega}^{-1}}\|_{2,1}, \tag{24}$$

where (a) comes from (21) and (22), (b) comes from (23), (c) comes from $\|(\widetilde{\beta}^l - \beta_*^l)_{\boldsymbol{\omega}^{-1}}\|_{2,1} \leq \|\widetilde{\beta}_{\boldsymbol{\omega}^{-1}}\|_{2,1} + \|\beta_{*\boldsymbol{\omega}^{-1}}\|_{2,1}$, (d) comes from $2\|\boldsymbol{X}\widetilde{\beta} - \boldsymbol{Y}\|_1 \geq \frac{5}{3}\|\boldsymbol{X}\widetilde{\beta} - \boldsymbol{Y}\|_1 \geq -\frac{5}{3}\|\boldsymbol{X}(\widetilde{\beta} - \beta_*)\|_1 + \frac{5}{3}\|\boldsymbol{\epsilon}\|_1$, (e) comes from the inequality that $\|\boldsymbol{X}(\widetilde{\beta} - \beta_*)\|_1 \leq \sqrt{n}\|\boldsymbol{X}(\widetilde{\beta} - \beta_*)\|_2$ and $\|\boldsymbol{X}\widetilde{\beta} - \boldsymbol{Y}\|_1 \leq \|\boldsymbol{X}(\widetilde{\beta} - \beta_*)\|_1 + \|\boldsymbol{\epsilon}\|_1$.

Since the inequality (24) is a second-order inequality of variable $\|\boldsymbol{X}(\widetilde{\beta} - \beta_*)\|_2/\sqrt{n}$, then the associate discriminant should be equal to or larger than 0, resulting in

$$\frac{1}{9}\delta^2(-11\|\widetilde{\beta}_{\boldsymbol{\omega}^{-1}}\|_{2,1}^2 + 66\|\beta_{*\boldsymbol{\omega}^{-1}}\|_{2,1}\|\widetilde{\beta}_{\boldsymbol{\omega}^{-1}}\|_{2,1} + 9\|\beta_{*\boldsymbol{\omega}^{-1}}\|_{2,1}^2)$$
$$+ \frac{\delta}{3n}\|\boldsymbol{\epsilon}\|_1(-2\|\widetilde{\beta}_{\boldsymbol{\omega}^{-1}}\|_{2,1} + 18\|\beta_{*\boldsymbol{\omega}^{-1}}\|_{2,1}) \geq 0,$$

from which we could conclude that

$$\|\widetilde{\beta}_{\boldsymbol{\omega}^{-1}}\|_{2,1} \leq 9\|\beta_{*\boldsymbol{\omega}^{-1}}\|_{2,1}.$$

$\square$

**Then, we proceed to prove Theorem 3.4.**

*Proof.* We write the objective function in (8) in the matrix norm and then have that

$$\frac{1}{2n}\|\boldsymbol{y} - \boldsymbol{X}\widetilde{\beta}\|_1^2 + \frac{1}{2}\delta^2\|\widetilde{\beta}_{\boldsymbol{\omega}^{-1}}\|_{2,1}^2 + \frac{\delta}{n}\|\boldsymbol{y} - \boldsymbol{X}\widetilde{\beta}\|\|\widetilde{\beta}_{\boldsymbol{\omega}^{-1}}\|_{2,1}$$
$$\leq \frac{1}{2n}\|\boldsymbol{y} - \boldsymbol{X}\beta_*\|_1^2 + \frac{1}{2}\delta^2\|\beta_{*\boldsymbol{\omega}^{-1}}\|_{2,1}^2 + \frac{\delta}{n}\|\boldsymbol{y} - \boldsymbol{X}\beta_*\|_1\|\beta_{*\boldsymbol{\omega}^{-1}}\|_{2,1}. \tag{25}$$

In this way, we have the following reformulation:

$$\frac{1}{2n}\|\boldsymbol{X}(\widetilde{\beta} - \beta_*)\|_2^2$$
$$\leq \frac{1}{n}\boldsymbol{\epsilon}^\top\boldsymbol{X}(\widetilde{\beta} - \beta_*) + \frac{\delta}{n}\|\boldsymbol{X}(\widetilde{\beta} - \beta_*)\|_1\|\widetilde{\beta}_{\boldsymbol{\omega}^{-1}}\|_{2,1} + \frac{\delta}{n}\|\boldsymbol{\epsilon}\|_1\left(\|\beta_{*\boldsymbol{\omega}^{-1}}\|_{2,1} - \|\widetilde{\beta}_{\boldsymbol{\omega}^{-1}}\|_{2,1}\right)$$
$$+ \frac{1}{2}\delta^2\|\beta_{*\boldsymbol{\omega}^{-1}}\|_{2,1}^2 - \frac{1}{2}\delta^2\|\widetilde{\beta}_{\boldsymbol{\omega}^{-1}}\|_{2,1}^2$$
$$\leq \frac{\delta}{2n}\|\boldsymbol{\epsilon}\|_1\|(\widetilde{\beta} - \beta_*)_{\boldsymbol{\omega}^{-1}}\|_{2,1} + \frac{\delta}{\sqrt{n}}\|\boldsymbol{X}(\widetilde{\beta} - \beta_*)\|_2\|\widetilde{\beta}_{\boldsymbol{\omega}^{-1}}\|_{2,1} + \frac{\delta}{n}\|\boldsymbol{\epsilon}\|_1\left(\|\beta_{*\boldsymbol{\omega}^{-1}}\|_{2,1} - \|\widetilde{\beta}_{\boldsymbol{\omega}^{-1}}\|_{2,1}\right)$$
$$+ \frac{1}{2}\delta^2\|\beta_{*\boldsymbol{\omega}^{-1}}\|_{2,1}^2 - \frac{1}{2}\delta^2\|\widetilde{\beta}_{\boldsymbol{\omega}^{-1}}\|_{2,1}^2, \tag{26}$$

where the last inequality comes from (23) and $\|\boldsymbol{X}(\widetilde{\beta} - \beta_*)\|_1 \leq \sqrt{n}\|\boldsymbol{X}(\widetilde{\beta} - \beta_*)\|_2$.

Similar to the proof of Theorem 2.3, we begin to give the upper bound of the prediction error. Two cases should be discussed.

**First Case:**
$$\|(\widetilde{\beta} - \beta_*)_{\boldsymbol{\omega}^{-1}}\|_{2,1} + 2\|\beta_{*\boldsymbol{\omega}^{-1}}\|_{2,1} - 2\|\widetilde{\beta}_{\boldsymbol{\omega}^{-1}}\|_{2,1} \leq 0. \tag{27}$$

In this case, we have that
$$\|\beta_{*\boldsymbol{\omega}^{-1}}\|_{2,1} - \|\widetilde{\beta}_{\boldsymbol{\omega}^{-1}}\|_{2,1} \leq \|(\widetilde{\beta} - \beta_*)_{\boldsymbol{\omega}^{-1}}\|_{2,1} + 2\|\beta_{*\boldsymbol{\omega}^{-1}}\|_{2,1} - 2\|\widetilde{\beta}_{\boldsymbol{\omega}^{-1}}\|_{2,1} \leq 0. \tag{28}$$

It follow from (26) that

$$\frac{1}{2n}\|\boldsymbol{X}(\widetilde{\beta} - \beta_*)\|_2^2$$
$$\leq \frac{\delta}{2n}\|\boldsymbol{\epsilon}\|_1\|(\widetilde{\beta} - \beta_*)_{\boldsymbol{\omega}^{-1}}\|_{2,1} + \frac{\delta}{\sqrt{n}}\|\boldsymbol{X}(\widetilde{\beta} - \beta_*)\|_2\|\widetilde{\beta}_{\boldsymbol{\omega}^{-1}}\|_{2,1} + \frac{\delta}{n}\|\boldsymbol{\epsilon}\|_1\left(\|\beta_{*\boldsymbol{\omega}^{-1}}\|_{2,1} - \|\widetilde{\beta}_{\boldsymbol{\omega}^{-1}}\|_{2,1}\right)$$
$$+ \frac{1}{2}\delta^2\|\beta_{*\boldsymbol{\omega}^{-1}}\|_{2,1}^2 - \frac{1}{2}\delta^2\|\widetilde{\beta}_{\boldsymbol{\omega}^{-1}}\|_{2,1}^2,$$
$$= \frac{\delta}{\sqrt{n}}\|\boldsymbol{X}(\widetilde{\beta} - \beta_*)\|_2\|\widetilde{\beta}_{\boldsymbol{\omega}^{-1}}\|_{2,1} + \frac{\delta}{2n}\|\boldsymbol{\epsilon}\|_1\left(\|(\widetilde{\beta} - \beta_*)_{\boldsymbol{\omega}^{-1}}\|_{2,1} + 2\|\beta_{*\boldsymbol{\omega}^{-1}}\|_{2,1} - 2\|\widetilde{\beta}_{\boldsymbol{\omega}^{-1}}\|_{2,1}\right)$$
$$+ \frac{1}{2}\delta^2\|\beta_{*\boldsymbol{\omega}^{-1}}\|_{2,1}^2 - \frac{1}{2}\delta^2\|\widetilde{\beta}_{\boldsymbol{\omega}^{-1}}\|_{2,1}^2,$$
$$\leq \frac{\delta}{\sqrt{n}}\|\boldsymbol{X}(\widetilde{\beta} - \beta_*)\|_2\|\widetilde{\beta}_{\boldsymbol{\omega}^{-1}}\|_{2,1}, \tag{29}$$

where the last inequality comes from (27) and (28).

Notice we have that

$$\|\beta_{*\boldsymbol{\omega}^{-1}}\|_{2,1} = \sum_{l \in J} \frac{1}{\omega_l} \|\beta_*^l\|_2 \le \sqrt{\sum_{l \in J} \frac{1}{\omega_l^2}} \|\beta_*\|_2. \tag{30}$$

Lemma E.1, (29) and (30) indicate that

$$\frac{1}{2n^2}\|\boldsymbol{X}(\widehat{\beta} - \beta_*)\|_2^2 \le 162\delta^2 \|\beta_{*\boldsymbol{\omega}^{-1}}\|_{2,1}^2 \le 162\delta^2 \sum_{l \in J} \frac{1}{\omega_l^2} \|\beta_*\|_2^2. \tag{31}$$

**Second Case:**
$$\|(\widetilde{\beta} - \beta_*)_{\boldsymbol{\omega}^{-1}}\|_{2,1} + 2\|\beta_{*\boldsymbol{\omega}^{-1}}\|_{2,1} - 2\|\widetilde{\beta}_{\boldsymbol{\omega}^{-1}}\|_{2,1} \ge 0.$$

Notice we have that

$$\|(\widetilde{\beta} - \beta_*)_{\boldsymbol{\omega}^{-1}}\|_{2,1} = \sum_{l=1}^{L} \frac{1}{\omega_l} \|\widetilde{\beta}^l - \beta_*^l\|_2 = \sum_{l \in J^c} \frac{1}{\omega_l} \|(\widetilde{\beta} - \beta_*)^l\|_2 + \sum_{l \in J} \frac{1}{\omega_l} \|(\widetilde{\beta} - \beta_*)^l\|_2,$$

$$\begin{aligned}
\|\beta_{*\boldsymbol{\omega}^{-1}}\|_{2,1} - \|\widetilde{\beta}_{\boldsymbol{\omega}^{-1}}\|_{2,1} &= \sum_{l \in J} \frac{1}{\omega_l} \|\beta_*^l\|_2 - \sum_{l=1}^{L} \frac{1}{\omega_l} \|\widetilde{\beta}^l\|_2 \\
&= \sum_{l \in J} \frac{1}{\omega_l} \|\beta_*^l\|_2 - \left( \sum_{l \in J} \frac{1}{\omega_l} \|\widetilde{\beta}^l\|_2 + \sum_{l \in J^c} \frac{1}{\omega_l} \|(\widetilde{\beta} - \beta_*)^l\|_2 \right) \\
&\le \sum_{l \in J} \frac{1}{\omega_l} \|(\widetilde{\beta} - \beta_*)^l\|_2 - \sum_{l \in J^c} \frac{1}{\omega_l} \|(\widetilde{\beta} - \beta_*)^l\|_2.
\end{aligned}$$

If we let $\widetilde{v} = \widetilde{\beta} - \beta_*$, then we have that

$$3 \sum_{l \in J} \frac{1}{\omega_l} \|\widetilde{v}^l\|_2 - \sum_{l \in J^c} \frac{1}{\omega_l} \|\widetilde{v}^l\|_2 \ge \|(\widetilde{\beta} - \beta_*)_{\boldsymbol{\omega}^{-1}}\|_{2,1} + 2\|\beta_{*\boldsymbol{\omega}^{-1}}\|_{2,1} - 2\|\widetilde{\beta}_{\boldsymbol{\omega}^{-1}}\|_{2,1} \ge 0, \tag{32}$$

indicating

$$\sum_{l \in J^c} \frac{1}{\omega_l} \|\widetilde{v}^l\|_2 \le 3 \sum_{l \in J} \frac{1}{\omega_l} \|\widetilde{v}^l\|_2.$$

In this way, the $\text{GRE}(g, 3)$ condition can be applied.

We also have the following from (26)

$$\begin{aligned}
&\frac{1}{2n} \|\boldsymbol{X}(\widetilde{\beta} - \beta_*)\|_2^2 \\
&\le \frac{\delta}{n} \|\boldsymbol{X}(\widetilde{\beta} - \beta_*)\|_1 \|\widetilde{\beta}_{\boldsymbol{\omega}^{-1}}\|_{2,1} + \frac{\delta}{2n} \|\boldsymbol{\epsilon}\|_1 \left( \|\beta_{*\boldsymbol{\omega}^{-1}}\|_{2,1} - \|\widetilde{\beta}_{\boldsymbol{\omega}^{-1}}\|_{2,1} + 2\|(\widetilde{\beta} - \beta_*)_{\boldsymbol{\omega}^{-1}}\|_{2,1} \right) \\
&\quad + \frac{1}{2}\delta^2 \|\beta_{*\boldsymbol{\omega}^{-1}}\|_{2,1}^2 - \frac{1}{2}\delta^2 \|\widetilde{\beta}_{\boldsymbol{\omega}^{-1}}\|_{2,1}^2 \\
&\overset{(a)}{\le} \frac{1}{4n} \|\boldsymbol{X}(\widetilde{\beta} - \beta_*)\|_2^2 + \frac{3\delta}{2n} \frac{\|\boldsymbol{\epsilon}\|_1}{n} \sum_{l \in J} \frac{1}{\omega_l} \|\widetilde{v}^l\|_2 + \frac{1}{2}\delta^2 \|\beta_{*\boldsymbol{\omega}^{-1}}\|_{2,1}^2 + \frac{1}{2}\delta^2 \|\widetilde{\beta}_{\boldsymbol{\omega}^{-1}}\|_{2,1}^2, \\
&\overset{(b)}{\le} \frac{1}{4n} \|\boldsymbol{X}(\widetilde{\beta} - \beta_*)\|_2^2 + \frac{3\delta}{2n} \frac{\|\boldsymbol{\epsilon}\|_1}{n} \sqrt{\sum_{l \in J} \frac{1}{\omega_l^2}} \|\widetilde{v}_{G_J}\|_2 + \frac{1}{2}\delta^2 \|\beta_{*\boldsymbol{\omega}^{-1}}\|_{2,1}^2 + \frac{1}{2}\delta^2 \|\widetilde{\beta}_{\boldsymbol{\omega}^{-1}}\|_{2,1}^2
\end{aligned}$$

where (a) comes from (32), and (b) comes from

$$\sum_{l \in J} \frac{1}{\omega_l} \|\widetilde{v}^l\|_2 \le \sqrt{\sum_{l \in J} \frac{1}{\omega_l^2}} \|\widetilde{v}_{G_J}\|_2.$$

It follows from $\text{GRE}(g, 3)$ and Lemma E.1 that

$$\frac{1}{4n}\|\boldsymbol{X}(\widetilde{\beta} - \beta_*)\|_2^2 \leq \frac{3\delta}{2\sqrt{n}}\sqrt{\sum_{l \in J}\frac{1}{\omega_l^2}}\frac{1}{\kappa(g, 3)}\frac{\|\boldsymbol{\epsilon}\|_1}{n}\|\boldsymbol{X}(\widetilde{\beta} - \beta_*)\|_2 + 41\delta^2\sum_{l \in J}\frac{1}{\omega_l^2}\|\beta_*\|_2^2.$$

Then, we could have that

$$\frac{1}{2n}\|\boldsymbol{X}(\widetilde{\beta} - \beta_*)\|_2^2 \leq 3\frac{\delta^2}{\kappa^2(g, 3)}\sum_{l \in J}\frac{1}{\omega_l^2}\max\left\{9\left(\frac{\|\boldsymbol{\epsilon}\|_1}{n}\right)^2, 164\kappa^2(g, 3)\|\beta_*\|_2^2\right\} \qquad (33)$$

Combining (31) and (33), we have that

$$\frac{1}{2n}\|\boldsymbol{X}(\widetilde{\beta} - \beta_*)\|_2^2 \leq 3\delta^2\sum_{l \in J}\frac{1}{\omega_l^2}\max\left\{\frac{9}{\kappa^2(g, 3)}\left(\frac{\|\boldsymbol{\epsilon}\|_1}{n}\right)^2, 164\|\beta_*\|_2^2\right\}.$$

$\square$

# F   Proof of Corollary 3.5

It follows from Lemma 3.1 in [17] that

$$\frac{2}{n}\|(\boldsymbol{X}^\top\boldsymbol{\epsilon})^l\|_2 \leq \frac{2\sigma}{\sqrt{n}}\sqrt{\text{tr}(\Psi_l) + 2\|\|\Psi_l\|\|(4\log L + \sqrt{2p_l\log L})}$$

holds with a probability greater than $1 - 2/L$, $\text{tr}(\Psi_l)$ denotes the trace of $\Psi_l$, and $\|\|\Psi_l\|\|$ denotes the maximum eigenvalue of $\Psi_l$. Since $\Psi_l = I_{p_l \times p_l}$, we have that $\|\|\Psi_l\|\| = 1$ and $\text{tr}(\Psi_l) = p_l$. Consequently, we have that

$$\frac{2}{n}\|(\boldsymbol{X}^\top\epsilon)^l\|_2 \leq \frac{2\sigma}{\sqrt{n}}\sqrt{p_l + 2(4\log L + \sqrt{2p_l\log L})} \leq \frac{2\sigma}{\sqrt{n}}\sqrt{3p_l + 9\log L},$$

holds with a probability greater than $1 - 2/L$.

Also, it follows from the proof of Corollary 2.4 that

$$\sigma\left(\sqrt{\frac{2}{\pi}} - \frac{1}{10}\right) \leq \frac{1}{n}\|\boldsymbol{\epsilon}\|_1$$

holds with a probability greater $1 - 2\exp\left(-C_1 n\right)$. Suppose we let

$$\frac{\delta}{\omega_l} = \frac{2}{\sqrt{\frac{2}{\pi}} - \frac{1}{10}}\sqrt{\frac{3p_l + 9\log L}{n}},$$

we have that

$$\frac{2\|(\boldsymbol{X}^\top\boldsymbol{\epsilon})^l\|_2}{\|\boldsymbol{\epsilon}\|_1} \leq \frac{\delta}{\omega_l}.$$

holds with a probability greater than $1 - 2\exp\left(-C_1 n\right) - 2/L$.

It follows from Theorem 3.4 that

$$\frac{1}{2n}\|\boldsymbol{X}(\widetilde{\beta} - \beta_*)\|_2^2 \leq 48\delta^2\sum_{l \in J}\frac{1}{\omega_l^2}\max\left\{\frac{9}{\kappa^2(g, 3)}\left(\frac{\|\boldsymbol{\epsilon}\|_1}{n}\right)^2, 164\|\beta_*\|_2^2\right\}$$

$$\leq 48\sum_{l \in J}\frac{3p_l + 9\log L}{n}\max\left\{\frac{9}{\kappa^2(s, 3)}\left(\frac{\|\boldsymbol{\epsilon}\|_1}{n}\right)^2, 164\|\beta_*\|_2^2\right\}$$

$$= 432\frac{|G_J| + g\log L}{n}\max\left\{\frac{9}{\kappa^2(s, 3)}\left(\frac{\|\boldsymbol{\epsilon}\|_1}{n}\right)^2, 164\|\beta_*\|_2^2\right\}.$$

# G   Proof of Corollary 3.7

Corollary 3.7 is straightforward due to Corollary 3.5 and the upper bound arguments in the proof in Corollary 2.6.

