# OpenReview forum: "High-dimensional (Group) Adversarial Training in Linear Regression"
_NeurIPS.cc/2024/Conference — NeurIPS 2024 poster_

### Official Review · Reviewer_fdf4 · 2024-06-24

**Soundness:** 3
**Presentation:** 3
**Contribution:** 2
**Rating:** 4
**Confidence:** 4

**Summary:**

This paper presents a non-asymptotic consistency analysis of prediction error for the adversarial training procedure under $l_\infty$ perturbation. It demonstrates that the convergence rate of the prediction error is up to a logarithmic factor. Additionally, the authors prove that the group adversarial training procedure achieves a superior upper bound on prediction error compared to classic adversarial training.

**Strengths:**

1. This paper applies the restricted eigenvalue condition and sparsity to deliver a convergence analysis, resulting in a better convergence rate.

2. The authors investigate the convergence rate of group adversarial training and achieve a faster upper bound for convergence.

**Weaknesses:**

1. The authors aim to connect their conclusions about a linear model to adversarial training, but adversarial training is a defense strategy commonly used in deep neural networks (DNN). The linear model is too simple and specific to accurately represent the behavior of adversarial training. The authors would benefit from studying adversarial training on a simple two-layer neural network or a convex function, not just the linear model.

2. The convergence rate for a linear model, as presented, is insufficient to illustrate the behavior of adversarial training effectively.


3. The points in Lines 142-145 lack supporting literature.

4. From Theorem 2.3 and Corollary 2.4, the authors derive a convergence rate of order $\frac{1}{n}$. Moreover, in Remark 2.8, the authors claim that the prediction error in [20] has a lower order $\frac{1}{\sqrt{n}}$. However, from Theorem 2 in [20], the convergence rate is also $\frac{1}{n}$ if $\delta\propto \frac{1}{\sqrt{n}}$ as your setting. The conclusion in this paper is very similar to that of [20]. While the restricted eigenvalue condition and sparsity might accelerate the convergence rate, I don’t see an improvement in this paper.

**Questions:**

See Weaknesses

---

> ### Author Rebuttal · Authors · 2024-08-01
>
> Thanks for the reviewer's comments on our work. We hope the following responses and clarifications can address the reviewer's concerns!
>
> **Comment 1**
> The authors aim to connect their conclusions about a linear model to adversarial training, but adversarial training is a defense strategy commonly used in deep neural networks (DNN). The linear model is too simple .... network or a convex function, not just the linear model.
>
> **Comment 2**
> The convergence rate for a linear model, as presented, is insufficient to illustrate the behavior of adversarial training effectively.
>
> **Response to Comment 1 and 2**
> Although the linear model seems simple, exploring the linear model is still common and crucial in the development of machine learning theories. Many existing works, e.g., [1-5], and our paper focus on the linear model for the adversarial training procedure. We explain the reasons as follows: Firstly, the linear model admits advanced analytical analysis. It will make the problem mathematically tractable and provide clear insights. For example, the minimax optimality for adversarial training under $\ell_\infty$-perturbation is proved in this paper, conveying the direct message that adversarial training under $\ell_\infty$-perturbation is statistically optimal. Moreover, the linear model could serve as an essential starting point for understanding more complex models. For example, it is well-known that the training dynamics of the wide neural network can be approximated by the linear model through the neural tangent kernel. Also, we want to emphasize that we are the first to prove the minimax optimality of the adversarial training in the linear model. We believe that this contribution has pushed the frontier of adversarial training theoretical exploration.
> We appreciate the reviewer’s consideration of this perspective and hope our clarifications underscore the rationale behind our focus on linear models and the significance of our contribution to the statistical theory of adversarial training.
>
> [1] A. Javanmard, M. Soltanolkotabi, and H. Hassani. Precise tradeoffs in adversarial training for linear regression. In Conference on Learning Theory, pages 2034–2078. PMLR, 2020
>
> [2] A. Ribeiro, D. Zachariah, F. Bach, and T. Schön. Regularization properties of adversarially-trained linear regression. Advances in Neural Information Processing Systems, 36, 2023
>
> [3] H. Taheri, R. Pedarsani, and C. Thrampoulidis. Asymptotic behavior of adversarial training in binary linear classification. IEEE Transactions on Neural Networks and Learning Systems, 2023
>
> [4] A. H. Ribeiro and T. B. Schön. Overparameterized linear regression under adversarial attacks. IEEE Transactions on Signal Processing, 71:601–614, 2023
>
> [5] E. Dobriban, H. Hassani, D. Hong, and A. Robey. Provable tradeoffs in adversarially robust classification. IEEE Transactions on Information Theory, 2023
>
>
>
> **Comment 3**
> The points in Lines 142-145 lack supporting literature.
>
> **Response to Comment 3**
> Line 142 - Line 145 are the analysis of Equation (3), which is a direct expansion of Equation (2). In the literature, this phenomenon has been referred to as the "regularization effect." We have mentioned the relevant literature in Line 146.
>
>
> **Comment 4**
> From Theorem 2.3 and Corollary 2.4, the authors derive a convergence rate of order $\frac{1}{n}$. Moreover, in Remark 2.8, the authors claim that the prediction error in [20] has a lower order $\frac{1}{\sqrt{n}}$. However, from Theorem 2 in [20], the convergence rate is also $\frac{1}{n}$ if $\delta \propto \frac{1}{\sqrt{n}}$ as your setting. The conclusion in this paper is very similar to that of [20]. While the restricted eigenvalue condition and sparsity might accelerate the convergence rate, I don't see an improvement in this paper.
>
> **Response to Comment 4**
> The rate derived in [20] is $\frac{1}{\sqrt{n}}$ rather than $\frac{1}{n}$. We would like to provide the following explanation: The upper bound of the error in [20] is given by  $8\delta\Vert\beta^\ast\Vert_1\left(\frac{1}{n}\Vert\varepsilon\Vert_1 + 10\delta \Vert\beta^\ast\Vert_1\right), $
> which can be written as:
> $$ 8\delta\Vert\beta^\ast\Vert_1\frac{1}{n}\Vert\varepsilon\Vert_1 + 80\delta^2\Vert\beta^\ast\Vert_1^2. \quad \quad \quad \quad \quad (1)$$ By choosing $\delta \propto \frac{1}{\sqrt{n}}$, the order of the first term in (1) is $\frac{1}{\sqrt{n}}$. A common misunderstanding regarding the order of $8\delta\Vert\beta^\ast\Vert_1\frac{1}{n}\Vert\varepsilon\Vert_1$ may come from the term $\frac{1}{n}\Vert\varepsilon\Vert_1$. At first glance, $\frac{1}{n}\Vert\varepsilon\Vert_1$seems to have the order $\frac{1}{n}$. However, since $\varepsilon$ is an $n$-dimensional vector, the resulting order of $ \Vert\varepsilon\Vert_1$ should be $n$, indicating the order of $\frac{1}{n}\Vert\varepsilon\Vert_1$ is $O(1)$ instead of $\frac{1}{n}$. Therefore, the order of the first term in (1) is $\frac{1}{\sqrt{n}}$. Consequently, the overall order of the error bound should be $\frac{1}{\sqrt{n}}$.
> For further clarification, please refer to the last paragraph on Page 8 in [20], where the authors claim that their order is $\frac{1}{\sqrt{n}}$:
> > "For $ \lambda \propto M \sigma \sqrt{(\log p) / n}$, we (with high probability) satisfy the condition in Theorem 3, obtaining: $\frac{1}{n}\Vert X(\widehat{\beta} -\beta^*)\Vert_2^2 \lesssim M \sigma \sqrt{(\log p) / n}$. For adversarial training, we can set: $\delta \propto M \sqrt{(\log p) / n}$, and (with high probability) satisfy the theorem condition, obtaining the same bound,"
>
> where the same bound denotes $M \sigma \sqrt{(\log p) / n}$, which has the order of $\frac{1}{\sqrt{n}}$.
>
> Given that our proved order is $ \frac{1}{n}$, **we believe our paper demonstrates an order improvement.**  We hope our explanations address the reviewer's concern and clarify the contribution of our work.

---

> ### Author Response · Authors · 2024-08-12
> **Gentle Reminder**
>
> Dear Reviewer,
>
> We hope this message finds you well. We are writing to kindly follow up on the rebuttal. We appreciate the effort you made to provide feedback on our paper. We would be grateful if you could let us know if our response has addressed your concerns.
>
>
>
> Thank you,
>
> Authors

---

### Official Review · Reviewer_J4Ez · 2024-07-08

**Soundness:** 4
**Presentation:** 4
**Contribution:** 3
**Rating:** 8
**Confidence:** 4

**Summary:**

The paper provides a high-dimensional analysis of Linear Regression in Adversarial training. It has two contributions:

1. it proves an improved convergence rate of prediction error of $1/n$  (previous work show $1/\sqrt{n}$).
2. It extends adversarial training for the group setting and extend the convergence results for it

**Strengths:**

The paper is well-written and clear.

The mathematical results seem to be consistent, and to the best of my knowledge are correct.

The author authors are very explicit about their contribution and present clear distinction from related work. Particularly, it is well contextualized and compared with [20], [26] and [28].

**Weaknesses:**

The numerical experiments are the main weakness. Not so many different configurations are tested. Still, since this is mostly a theoretical paper, I don't see this as a major problem. But I believe studying it for different settings could strength the paper

I think the presentation of the numerical results could be somehow improved,
1. Figure 1 could be interesting to have confidence intervals and also have in the plot the final values that the coefficients converge to.
2. Figure 2 it would be better to have log 10 base, it is a bit hard to read with ln.

**Questions:**

It is unclear to me why in the numerical experiments what constant is used when setting the perturbation proportional to $1/\sqrt{n}$? How is it chosen, because I imagine is hard to guarantee in practice that the conditions of Theorem 2.3 are being satisfied.

**Limitations:**

The paper has no clear societal impact. See weaknesses.

---

> ### Author Rebuttal · Authors · 2024-08-04
>
> Great thanks for the reviewer's appreciation of our work! Regarding the numerical experimental improvement, please see our response below and the revised figures in the pdf file attached in the global response.
>
>
> **Comment 1**:
> The numerical experiments are the main weakness. Not so many different configurations are tested. Still, since this is mostly a theoretical paper, I don't see this as a major problem. But I believe studying it for different settings could strength the paper.
>
> **Response to Comment 1**:
> We appreciate your recognition of the theoretical contributions of the paper. We acknowledge that the numerical experiments could benefit from a wider variety of configurations.
> Due to the page limit of the rebuttal attachments,
> we will include the results of additional settings in future revisions.
>
>
> **Comment 2**:
> I think the presentation of the numerical results could be somehow improved,
> Figure 1 could be interesting to have confidence intervals and also have in the plot the final values that the coefficients converge to.
> Figure 2 it would be better to have log 10 base, it is a bit hard to read with ln.
>
> **Response to Comment 2**:
> Thanks for the suggestions for improving the presentations of our numerical results. We have added the confidence interval, indicated the values of the coefficient converging, and changed to $\log 10$ base in the figures. Please see the pdf file with the revised figures attached in the global response.
>
> We have some explanations of the figures as follows.
>
>
> We plot the curve of $log_{10}$(prediction error) versus $log_{10}$(sample size)  with error bars. We can observe that the slopes of two curves are
>  approximately equal to $-1$, which is consistent with our theoretical analysis, where we have proved that the prediction error for high-dimensional (group) adversarial training is of the order $1/n$. Further,
> the curve and error bar of group adversarial training are below those of adversarial training, indicating the superiority of group adversarial training.
>
> We also plot the coefficient estimation paths of (group) adversarial training with error bars. Both adversarial training and group adversarial training can shrink the parameter estimation, while group adversarial training performs a better shrinkage effect,
> In addition, the final values that the coefficients converge to are annotated in the figures. Given the ground-truth non-zero values [0.1, 0.15, 0.2, 0.25, 0.9, 0.95, 1, 1.05], the final values of group adversarial training are closer to the ground-truth, indicating that the group adversarial training output a more accurate estimation.
>
> **Comment 3**:
> It is unclear to me why in the numerical experiments what constant is used when setting the perturbation proportional to $1/\sqrt{n}$
> ? How is it chosen, because I imagine is hard to guarantee in practice that the conditions of Theorem 2.3 are being satisfied.
>
> **Response to Comment 3**:
> The reviewer is correct that it is hard to guarantee the conditions in Theorem 2.3 in practice.
> As noted, we choose the order $1/\sqrt{n}$ because Corollary 2.4 shows that the minimax convergence rate can be achieved if the perturbation magnitude $\delta$ is of this order. This order is also recommended in the literature [26] (reference in the paper). For the constant, we selected $1$ for simplicity and experimental convenience. This constant allows us to illustrate the theoretical results effectively without complicating the numerical setup. We will clarify this setting clearly in our future revisions.

---

> ### Comment · Reviewer_J4Ez · 2024-08-08
>
> I read the other reviews and comments. I believe the focus on linear models is interesting and well-justified and I don't think other scores should penalize the paper for it.  The paper is good in what it does and has a clear contribution.
>
> I thank the reviewer for addressing my concerns and I raise my score to strong accept.

---

> > ### Author Response · Authors · 2024-08-09
> >
> > Thank you for recognizing our contributions and the value of our focus on linear models! We truly appreciate your positive evaluations and are greatly encouraged by your decision to raise your score to a strong accept.

---

### Official Review · Reviewer_6XeH · 2024-07-12

**Soundness:** 3
**Presentation:** 3
**Contribution:** 2
**Rating:** 5
**Confidence:** 3

**Summary:**

This paper provided an theoretical analysis for the optimality of adversarial training methods on linear regression, and further explored the advantages of the group adversarial training method, comparing with general adversarial training method. There are also experiments supporting their points.

**Strengths:**

1.The paper is well-organized with clear statement. The contributions are stated clearly, and it is easy to follow the logic and flow of the paper.

2.The theoretical results are discussed detailedly, which helps to understand the theorems.

**Weaknesses:**

1.As the paper is mainly focused on empirical errors, the contributions seem to be not enough. It is better to extend the results on test error analysis, which may be more interesting.

2.The tightness of such upper bounds in main theorems has not been proved.

3.The linear model seems to be quite constrained, it is recommended to extend to random feature models or other neural network models, such as two-layer NTK or diagonal network.

**Questions:**

See weakness.

**Limitations:**

No negative social impact.

---

> ### Author Rebuttal · Authors · 2024-08-01
>
> Thanks for the reviewer's insightful comments. We hope our response and clarifications can address the reviewer's concerns!
>
> **Comment 1:** As the paper is mainly focused on empirical errors, the contributions seem to be not enough. It is better to extend the results on test error analysis, which may be more interesting.
>
> **Response to Comment 1:**
> Thanks for the reviewer for pointing out this. We clarify the error framework used in this paper as follows.
>
>
> 1.
> We are using the prediction error, $\frac{1}{n}\Vert \mathbf{X}(\hat{\beta}-\beta^\ast) \Vert_2^2$ ,instead of the empirical error.
> In the framework of the non-asymptotic high-dimensional statistical analysis, the prediction error is typically used, seeing [1-4]. The prediction error can help us directly quantify the derivation in $\hat{\beta}$ from $\beta^\ast$.
>
>
> 2. The prediction error, $\frac{1}{n}\Vert \mathbf{X}(\hat{\beta}-\beta^\ast) \Vert_2^2$, is also called 'in-sample' prediction error. The 'test error' mentioned by the reviewer may be referred to as the 'out-of-sample' prediction error. We provide explanations for why the in-sample prediction error is usually preferred in high-dimensional analysis as follows.
> Firstly, high-dimensional settings typically involve situations where the number of input variables is much larger than the number of observations. In such cases, splitting the data into training and test sets can result in very few observations in the test set, making out-of-sample prediction errors unreliable.
> Secondly, the in-sample prediction error will enable the application of the concentration inequality, resulting in explicit probabilistic bounds analysis. Thus, in-sample prediction error can admit advanced theoretical analysis.
>
>
> [1] Wainwright, Martin J. High-dimensional statistics: A non-asymptotic viewpoint. Vol. 48. Cambridge university press, 20
>
> [2] G. Raskutti, M. J. Wainwright, and B. Yu. Minimax rates of estimation for high-dimensional linear regression over lq-balls. IEEE Transactions on information theory, 57(10):6976–6994,
> 373 2011.
>
> [3] P. C. Bellec, G. Lecué, and A. B. Tsybakov. Slope meets lasso: improved oracle bounds and optimality. The Annals of Statistics, 46(6B):3603–3642, 2018.
>
> [4] K. Lounici, M. Pontil, S. van de Geer, and A. B. Tsybakov. Oracle inequalities and optimal inference under group sparsity. The Annals of Statistics, pages 2164–2204, 2011.
>
> **Comment 2:** The tightness of such upper bounds in main theorems has not been proved.
>
> **Response to Comment 2:**  Thank you for your comment regarding the tightness of the upper bounds in our main theorems. We appreciate the opportunity to clarify this point.
> The upper bounds in our main theorems are indeed tight, as we have demonstrated that the error of the adversarial training can achieve the minimax rate. Specifically, the error order in our paper is $s\log p/n$, which matches the minimax lower bound given in [2,19] (the references in our paper), seeing Line 206-208. We hope this explanation addresses your concern.
>
>
>
>
> **Comment 3:** The linear model seems to be quite constrained, it is recommended to extend to random feature models or other neural network models, such as two-layer NTK or diagonal network.
>
> **Response to Comment 3:**
> Thanks for the reviewer's valuable suggestions on possible extensions.
> The reason we focus on the linear model is that it will allow advanced analytical analysis.
> It will make the problem mathematically tractable and provide clear insights.
> For example, this paper proves the minimax optimality for adversarial training under $\ell_\infty$-perturbation, conveying the direct message that adversarial training is statistically optimal.
> Moreover, the linear model could serve as an essential starting point for understanding more complex models. For example, it is well-known that training dynamics of wide neural network can be approximated by the linear model through neural tangent kernel.
> We appreciate your suggestions for the extensions.
> The extensions to the more complex models are very promising but may require intensive additional proof work, so we will consider the extensions seriously in our future work.

---

> ### Author Response · Authors · 2024-08-12
> **Gentle Reminder**
>
> Dear Reviewer,
>
> We hope this message finds you well. We are writing to kindly follow up on the rebuttal. We appreciate the effort you made to provide feedback on our paper. We would be grateful if you could let us know if our response has addressed your concerns.
>
> Thank you,
>
> Authors

---

> > ### Comment · Reviewer_6XeH · 2024-08-13
> > **Response to the rebuttal**
> >
> > Many thanks for the authors' response. This paper has an interesting story, but maybe it need to be polished from my side. I will maintain my score.

---

### Official Review · Reviewer_CJZh · 2024-07-12

**Soundness:** 2
**Presentation:** 2
**Contribution:** 1
**Rating:** 4
**Confidence:** 3

**Summary:**

The paper studies adversarial training in high-dimensional linear regression under $\ell_\infty$-perturbations and group adversarial training. The paper also provides non-asymptotic consistency analysis.

**Strengths:**

The associated convergence rate of prediction error achieves a minimax rate up to a logarithmic factor. The authors also claim that group adversarial training offers a better prediction error upper bound under certain group-sparsity patterns.

**Weaknesses:**

- Remark 2.8: The authors say that they improve the convergence rate of the prediction error from $\frac{1}{\sqrt{n}}$ to $\frac{1}{n}$. Is the improvement possible only due to the additional assumption of restricted eigenvalue condition and sparsity information? If so, it should be clearly stated in the abstract and introduction. Otherwise, it gives the wrong impression or over-claims that the improvement is achieved without any additional assumptions.

- Is the weight vector $\mathbf{w}$ defined in line 234 assumed to be known in group adversarial training? If so, it should be stated clearly as an assumption. If not, does an algorithm exist to estimate it? The results are obviously heavily dependent on this parameter.

- The abstract can be improved by including more technical information about the specific contributions. For example, is the analysis improving any rates as compared to the existing literature? If so, what specific theoretical analysis or assumption helped to achieve that improvement? For example, the last line in the abstract mentions certain group sparsity patterns. It would be helpful if the authors could explain more about this particular sparsity pattern so that a reader gets an understanding of whether the requirement is restrictive or easily achievable.

- Line 30: The authors could probably clarify with more technical details why existing literature thinks that $\ell_\infty$-perturbations can be helpful in recovering sparsity.

- Line 87-97: It is repetitive.

- Line 229 has typo: $s = \infty$  should be used in underscore.

**Questions:**

- The proof of Corollary 2.4 uses Gaussian tail bounds. Can the proof be generalized for sub-Gaussian distribution?

- Are the bounds derived in Corollary 3.5 or 3.6 tight? Meaning, can the authors show the equivalence to the bounds derived in Corollary  2.4 or 2.5 by making the number of groups = 1 in Corollary 3.5 or 3.6?

- Line 55: what is the sparsity condition exactly?

- Line 65: What is the certain group sparsity structures exactly?

**Limitations:**

- At the end of Section 3, the authors claim that the inferences made are similar to the differences between Lasso and group Lasso. If that is the case, what are the new insights that the proposed theoretical analysis is bringing in? It seems like a paper that rigorously verifies the expected results, which have been well explored in the literature for similar methods like Lasso and group Lasso.

- The equality assumption on $\delta$ or $\frac{\delta}{w_l}$ in Corollary 2.4 or Corollary 3.5, respectively, seem quite restrictive. Is the theoretical analysis not useful for any other $\delta$? Any form of upper bound or lower bound on $\delta$ could have been more helpful.

- If I understand correctly, the paper analyzes the particular case of $r = 2$ and $s = \infty$ only. But the problem is defined for general $r$ and $s$ in proposition 3.1 and equation after line 233. It gives the wrong impression to the reader that the problem is analyzed for general $r$ and $s$.

---

> ### Author Rebuttal · Authors · 2024-08-03
>
> Thanks for the reviewer's careful and detailed comments on our work. We hope our responses and clarifications can address the reviewer's concerns!
>
> **Comment 1** ...improve the convergence rate......clearly stated in the abstract...
>
> **Response**
> The reviewer is correct that the order improvement is based on the restricted eigenvalue condition, a standard assumption in the literature of sparse high-dimensional analysis. We also assume that the ground truth parameter $\beta_\ast\in\mathbb{R}^p$ is supported on a subset of {1,...,p}. We will revise the abstract and introduction to state these assumptions more clearly.
>
> **Comment 2** Is the weight vector $\mathbf{w}$ defined in line 234 assumed to be known in group adversarial training?...
>
> **Response** $\mathbf{w}$ is a hyperparameter and does not need to be estimated. Additionally, in the order analysis presented in Corollary 3.5, the error upper bound no longer depends on $\mathbf{w}$.
>
> **Comment 3** The abstract can be improved by including more technical information about the specific contributions...
>
> **Response** The main contributions in this paper are:  (a) We are the first to show that adversarial training can achieve minimax optimality. We prove this under the restricted eigenvalue condition. (b) We analyze the group adversarial training and prove that it enjoys a better prediction error bound when group sparsity patterns are known.  The group sparsity pattern means that the variables act in groups, and sparsity exists at the level of groups of variables instead of individual variables. The group patterns exist in many real-world problems. For example, groups of genes act together in pathways in gene-expression arrays. Also, if an input variable is a multilevel factor and dummy variables are introduced, these dummy variables act in a group. We will make all these assumptions and contributions clearer in the abstract.
>
> **Comment 4** The authors...why existing literature thinks that $\ell_{\infty}$-perturbations can be helpful in recovering sparsity.
>
> **Response** [26] has proved that the asymptotic distribution of adversarial training estimator under $\ell_\infty$-perturbation has a positive mass at $0$ when the underlying parameter is $0$. Other types of perturbation do not have this property. We will state these technical details carefully in our revision.
>
> **Comment 5 and 6** Line 87-97: It is repetitive. Line 229 has typo...
>
> **Response:**
> We will revise these lines.
>
>
> **Comment 7**  ...Can the proof be generalized for sub-Gaussian distribution?
>
> **Response**  Corollary 2.4 can be generalized to sub-Gaussian distributions since sub-Gaussian distributions share similar tail behavior with Gaussian distributions. We will revise the proof to include the generalization for sub-Gaussian distributions.
>
> **Comment 8** Are the bounds derived in Corollary 3.5 or 3.6 tight?..
>
> **Response:** Thanks for pointing this out. The bounds derived in Corollary 3.5 or 3.6 are tight. If we make the number of groups equal to $p$, i.e., each group only has one component, then we will have that $L=p,p_l=1, \vert G_J\vert=g$. The resulting error bound is $g\log p/n$, where $g$ denotes the number of nonzero components of $\beta_\ast$. This order matches what is derived in Corollary 2.4 or 2.5. We will conclude these as a remark under Corollary 3.5 and 3.6.
>
> **Comment 9** what is the sparsity condition exactly?
>
> **Response** The sparsity condition means that the number of non-zero coefficients is less than the total number of coefficients, i.e., the ground truth $\beta_\ast\in\mathbb{R}^p$ is supported on a subset of {1,...,p}. We will clarify this definition in the revision.
>
> **Comment 10** What is the certain group sparsity structures exactly?
>
> **Response** The group sparsity structure means that sparsity is enforced at the group rather than at the individual level.
> Specifically, suppose the index set {1,...,p} has the prescribed (disjoint) partition {1,...,p}$=\bigcup_{l=1}^L G_l$.
> $J\subset$ {1,..., L} denotes a set of groups and $\beta_\ast\in\mathbb{R}^p$ is supported at these J groups, i.e., $\beta_\ast$ is supported on the  $G_J=\bigcup _{l\in J}G_l$.
>
> **Comment 11** ...the inferences made are similar to the differences between Lasso and group Lasso..what are the new insights that...
>
> **Response** Our work focuses on the adversarial training problem, which is inherently different from the Lasso problem, and our contributions are (1) the adversarial training is minimax optimal, and (2) group adversarial training can improve the error bound. These conclusions have never been explored in the literature. We mention and correlate adversarial training with LASSO because both LASSO and $\ell_\infty$-perturbed adversarial training could recover sparsity and achieve the minimax convergence rate. In this way, it is not surprising that the theoretical error bounds of (group) adversarial training are consistent with (group) LASSO. We will add these discussions to avoid confusion.
>
> **Comment 12** The equality assumption on $\delta$ or $\frac{\delta}{w_l}$ in Corollary 2.4 or Corollary 3.5, respectively, seem quite restrictive...
>
> **Response** $\delta$ should be in the order of $1/\sqrt{n}$ to make the corollaries hold due to the structure of the concentration inequality of Gaussian distribution. We admit that this setting seems to be restrictive. But we would like to interpret this result as a recommendation of order choice. That is to say, the order of $1/\sqrt{n}$ is recommended in order to achieve the fast convergence rate. This order choice is also recommended in the literature [26] to achieve sparsity, seeing Remark 2.5 in our paper.
>
>
> **Comment 13** If I understand correctly, the paper analyzes ... gives the wrong impression to the reader that the problem is analyzed for general $r$ and $s$.
>
> **Response** The reviewer is correct. Thanks for pointing this out. We will clarify the scope of our analysis more clearly.

---

> ### Author Response · Authors · 2024-08-12
> **Gentle Reminder**
>
> Dear Reviewer,
>
> We hope this message finds you well. We are writing to kindly follow up on the rebuttal. We appreciate the effort you made to provide feedback on our paper. We would be grateful if you could let us know if our response has addressed your concerns.
>
> Thank you,
>
> Authors

---

> > ### Comment · Reviewer_CJZh · 2024-08-12
> >
> > I thank the authors for their detailed response. I will maintain my original score.

---

### Author Rebuttal · Authors · 2024-08-04

We have revised the figures for the numerical experiments as requested by Reviewer J4Ez. Please see the attached pdf file.

---

### Decision · Program_Chairs · 2024-09-25

**Decision:**

Accept (poster)

**Comment:**

This paper mathematically studies sparse linear regression under adversarial training.  I am happy to recommend acceptance, however I request that the authors pay close attention to the many comments and criticisms stated by the reviewers; for example, as a purely presentation issue discussed heavily by the reviewers, it is very odd to give the rate as $1/n$ rather than $s \ln p / n$ which is the hallmark of sparse linear regression.  Thank you!